# Denoising Diffusion Restoration Models

**Bahjat Kawar**
Department of Computer Science
Technion, Haifa, Israel
bahjat.kawar@cs.technion.ac.il

**Michael Elad**
Department of Computer Science
Technion, Haifa, Israel
elad@cs.technion.ac.il

**Stefano Ermon**
Department of Computer Science
Stanford, California, USA
ermon@cs.stanford.edu

**Jiaming Song**
NVIDIA
Santa Clara, California, USA
jiamings@nvidia.com

## Abstract

Many interesting tasks in image restoration can be cast as linear inverse problems. A recent family of approaches for solving these problems uses stochastic algorithms that sample from the posterior distribution of natural images given the measurements. However, efficient solutions often require problem-specific supervised training to model the posterior, whereas unsupervised methods that are not problem-specific typically rely on inefficient iterative methods. This work addresses these issues by introducing Denoising Diffusion Restoration Models (DDRM), an efficient, unsupervised posterior sampling method. Motivated by variational inference, DDRM takes advantage of a pre-trained denoising diffusion generative model for solving any linear inverse problem. We demonstrate DDRM's versatility on several image datasets for super-resolution, deblurring, inpainting, and colorization under various amounts of measurement noise. DDRM outperforms the current leading unsupervised methods on the diverse ImageNet dataset in reconstruction quality, perceptual quality, and runtime, being $5\times$ faster than the nearest competitor. DDRM also generalizes well for natural images out of the distribution of the observed ImageNet training set.[1]

## 1 Introduction

Many problems in image processing, including super-resolution [31, 17], deblurring [28, 48], inpainting [55], colorization [29, 58], and compressive sensing [1], are instances of linear inverse problems, where the goal is to recover an image from potentially noisy measurements given through a known linear degradation model. For a specific degradation model, image restoration can be addressed through end-to-end *supervised* training of neural networks, using pairs of original and degraded images [14, 58, 41]. However, real-world applications such as medical imaging often require flexibility to cope with multiple, possibly infinite, degradation models [46]. Here, *unsupervised* approaches based on learned priors [36], where the degradation model is only known and used during inference, may be more desirable since they can adapt to the given problem without re-training [51]. By learning sound assumptions over the underlying structure of images (*e.g.*, priors, proximal operators or denoisers), unsupervised approaches can achieve effective restoration without training on specific degradation models [51, 40].

Under this unsupervised setting, priors based on deep neural networks have demonstrated impressive empirical results in various image restoration tasks [40, 50, 43, 38, 15]. To recover the signal,

---

[1]Project website: https://ddrm-ml.github.io/

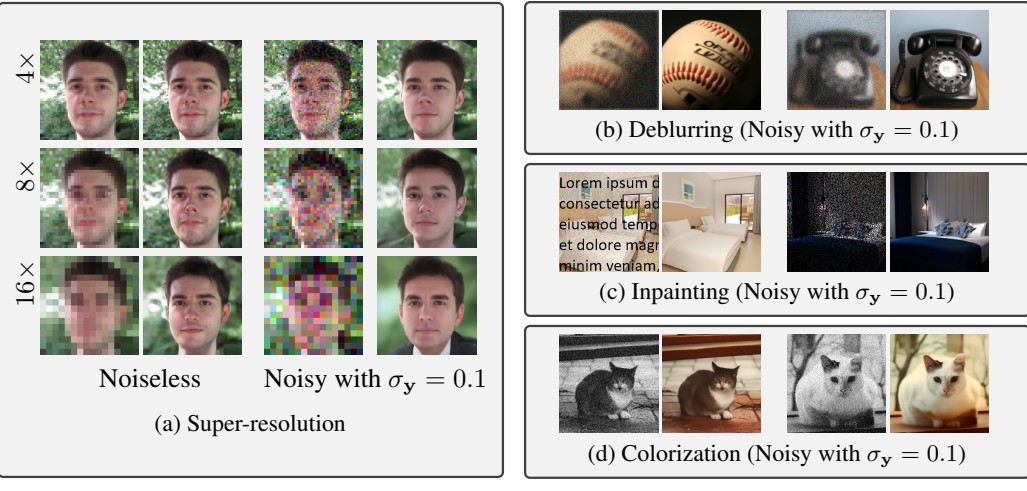

Figure 1: Pairs of measurements and recovered images with a 20-step DDRM on super-resolution, deblurring, inpainting, and colorization, with or without noise, and with unconditional generative models. The images are not accessed during training.

most existing methods obtain a prior-related term over the signal from a neural network (*e.g.*, the distribution of natural images), and a likelihood term from the degradation model. They combine the two terms to form a posterior over the signal, and the inverse problem can be posed as solving an optimization problem (*e.g.*, maximum a posteriori [8, 40]) or solving a sampling problem (*e.g.*, posterior sampling [2, 3, 25]). Then, these problems are often solved with iterative methods, such as gradient descent or Langevin dynamics, which may be demanding in computation and sensitive to hyperparameter tuning. An extreme example is found in [30] where a "fast" version of the algorithm uses 15,000 neural function evaluations (NFEs).

Inspired by this unsupervised line of work, we introduce an efficient approach named Denoising Diffusion Restoration Models (DDRM), that can achieve competitive results in as low as 20 NFEs. DDRM is a denoising diffusion generative model [44, 19, 45] that gradually and stochastically denoises a sample to the desired output, conditioned on the measurements and the inverse problem. This way we introduce a variational inference objective for learning the posterior distribution of the inverse problem at hand. We then show its equivalence to the objective of an unconditional denoising diffusion generative model [19], which enables us to deploy such models in DDRM for various linear inverse problems (see Figure 2). To our best knowledge, DDRM is the first general sampling-based inverse problem solver that can efficiently produce a range of high-quality, diverse, yet valid solutions for general content images.

We demonstrate the empirical effectiveness of DDRM by comparing with various competitive methods based on learned priors, such as Deep Generative Prior (DGP) [38], SNIPS [25], and Regularization by Denoising (RED) [40]. On ImageNet examples, DDRM mostly outperforms the neural network baselines under noiseless super-resolution and deblurring measured in PSNR and KID [5], and is at least $50\times$ more efficient in terms of NFEs when it is second-best. Our advantage becomes even larger when measurement noise is involved, as noisy artifacts produced by iterative methods do not appear in our case. Over various real-world images, we further show DDRM results on super-resolution, deblurring, inpainting and colorization (see Figure 1). A DDRM trained on ImageNet also works on images that are out of its training set distribution (see Figure 6).

## 2 Background

**Linear Inverse Problems.** A general linear inverse problem is posed as

$$\mathbf{y} = \boldsymbol{H}\mathbf{x} + \mathbf{z}, \tag{1}$$

where we aim to recover the signal $\mathbf{x} \in \mathbb{R}^n$ from measurements $\mathbf{y} \in \mathbb{R}^m$, where $\boldsymbol{H} \in \mathbb{R}^{m\times n}$ is a known linear degradation matrix, and $\mathbf{z} \sim \mathcal{N}(0, \sigma_{\mathbf{y}}^2 \boldsymbol{I})$ is an *i.i.d.* additive Gaussian noise with known variance. The underlying structure of $\mathbf{x}$ can be represented via a generative model, denoted

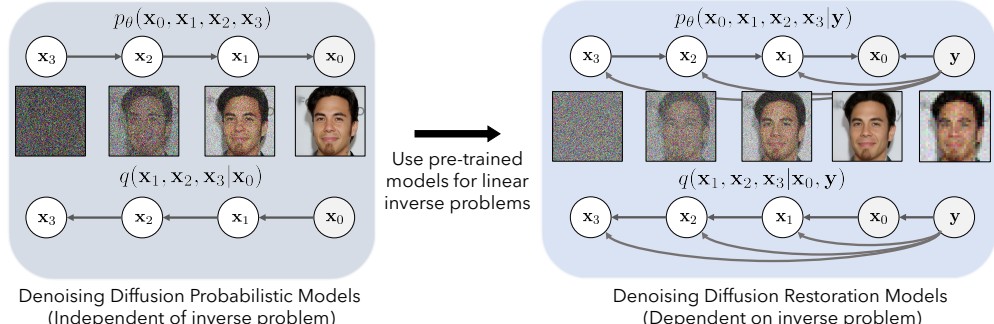

Figure 2: Illustration of our DDRM method for a specific inverse problem (super-resolution + denoising). We can use unsupervised DDPM models as a good solution to the DDRM objective.

as $p_\theta(\mathbf{x})$. Given $\mathbf{y}$ and $\boldsymbol{H}$, a posterior over the signal can be posed as: $p_\theta(\mathbf{x}|\mathbf{y}) \propto p_\theta(\mathbf{x})p(\mathbf{y}|\mathbf{x})$, where the "likelihood" term $p(\mathbf{y}|\mathbf{x})$ is defined via Equation (1); such an approach leverages a learned prior $p_\theta(\mathbf{x})$, and we call it an "unsupervised" approach based on the terminology in [36], as the prior does not necessarily depend on the inverse problem. Recovering $\mathbf{x}$ can be done by sampling from this posterior [2], which may require many iterations to produce a good sample. Alternatively, one can also approximate this posterior by learning a model via amortized inference (*i.e.*, supervised learning); the model learns to predict $\mathbf{x}$ given $\mathbf{y}$, generated from $\mathbf{x}$ and a specific $\boldsymbol{H}$. While this can be more efficient than sampling-based methods, it may generalize poorly to inverse problems that have not been trained on.

**Denoising Diffusion Probabilistic Models.** Structures learned by generative models have been applied to various inverse problems and often outperform data-independent structural constraints such as sparsity [7]. These generative models learn a model distribution $p_\theta(\mathbf{x})$ that approximates a data distribution $q(\mathbf{x})$ from samples. In particular, diffusion models have demonstrated impressive unconditional generative modeling performance on images [13]. Diffusion models are generative models with a Markov chain structure $\mathbf{x}_T \rightarrow \mathbf{x}_{T-1} \rightarrow \ldots \rightarrow \mathbf{x}_1 \rightarrow \mathbf{x}_0$ (where $\mathbf{x}_t \in \mathbb{R}^n$), which has the following joint distribution:

$$p_\theta(\mathbf{x}_{0:T}) = p_\theta^{(T)}(\mathbf{x}_T) \prod_{t=0}^{T-1} p_\theta^{(t)}(\mathbf{x}_t|\mathbf{x}_{t+1}).$$

After drawing $\mathbf{x}_{0:T}$, only $\mathbf{x}_0$ is kept as the sample of the generative model. To train a diffusion model, a fixed, factorized variational inference distribution is introduced:

$$q(\mathbf{x}_{1:T}|\mathbf{x}_0) = q^{(T)}(\mathbf{x}_T|\mathbf{x}_0) \prod_{t=0}^{T-1} q^{(t)}(\mathbf{x}_t|\mathbf{x}_{t+1}, \mathbf{x}_0),$$

which leads to an evidence lower bound (ELBO) on the maximum likelihood objective [44]. A special property of some diffusion models is that both $p_\theta^{(t)}$ and $q^{(t)}$ are chosen as conditional Gaussian distributions for all $t < T$, and that $q(\mathbf{x}_t|\mathbf{x}_0)$ is also a Gaussian with known mean and covariance, *i.e.*, $\mathbf{x}_t$ can be treated as $\mathbf{x}_0$ directly corrupted with Gaussian noise. Thus, the ELBO objective can be reduced into the following denoising autoencoder objective (please refer to [45] for derivations):

$$\sum_{t=1}^{T} \gamma_t \mathbb{E}_{(\mathbf{x}_0, \mathbf{x}_t) \sim q(\mathbf{x}_0)q(\mathbf{x}_t|\mathbf{x}_0)} \left[ \|\mathbf{x}_0 - f_\theta^{(t)}(\mathbf{x}_t)\|_2^2 \right] \tag{2}$$

where $f_\theta^{(t)}$ is a $\theta$-parameterized neural network that aims to recover a noiseless observation from a noisy $\mathbf{x}_t$, and $\gamma_{1:T}$ are a set of positive coefficients that depend on $q(\mathbf{x}_{1:T}|\mathbf{x}_0)$.

## 3 Denoising Diffusion Restoration Models

Inverse problem solvers based on posterior sampling often face a dilemma: unsupervised approaches apply to general problems but are inefficient, whereas supervised ones are efficient but can only address specific problems.

To solve this dilemma, we introduce Denoising Diffusion Restoration Models (DDRM), an unsupervised solver for general linear inverse problems, capable of handling such tasks with or without noise in the measurements. DDRM is efficient and exhibits competitive performance compared to popular unsupervised solvers [40, 38, 25].

The key idea behind DDRM is to find an unsupervised solution that also suits supervised learning objectives. First, we describe the variational objective for DDRM over a specific inverse problem (Section 3.1). Next, we introduce specific forms of DDRM that are suitable for linear inverse problems and allow pre-trained unconditional and class-conditional diffusion models to be used directly (Sections 3.2, 3.3). Finally, we discuss practical algorithms that are compute and memory efficient (Sections 3.4, 3.5).

## 3.1 Variational Objective for DDRM

For any linear inverse problem, we define DDRM as a Markov chain $\mathbf{x}_T \to \mathbf{x}_{T-1} \to \ldots \to \mathbf{x}_1 \to \mathbf{x}_0$ conditioned on $\mathbf{y}$, where

$$p_\theta(\mathbf{x}_{0:T}|\mathbf{y}) = p_\theta^{(T)}(\mathbf{x}_T|\mathbf{y}) \prod_{t=0}^{T-1} p_\theta^{(t)}(\mathbf{x}_t|\mathbf{x}_{t+1}, \mathbf{y})$$

and $\mathbf{x}_0$ is the final diffusion output. In order to perform inference, we consider the following factorized variational distribution conditioned on $\mathbf{y}$:

$$q(\mathbf{x}_{1:T}|\mathbf{x}_0, \mathbf{y}) = q^{(T)}(\mathbf{x}_T|\mathbf{x}_0, \mathbf{y}) \prod_{t=0}^{T-1} q^{(t)}(\mathbf{x}_t|\mathbf{x}_{t+1}, \mathbf{x}_0, \mathbf{y}),$$

leading to an ELBO objective for diffusion models conditioned on $\mathbf{y}$ (details in Appendix A).

In the remainder of the section, we construct suitable variational problems given $\boldsymbol{H}$ and $\sigma_{\mathbf{y}}$ and connect them to unconditional diffusion generative models. To simplify notations, we will construct the variational distribution $q$ such that $q(\mathbf{x}_t|\mathbf{x}_0) = \mathcal{N}(\mathbf{x}_0, \sigma_t^2 \boldsymbol{I})$ for noise levels $0 = \sigma_0 < \sigma_1 < \sigma_2 < \ldots < \sigma_T$.[2] In Appendix B, we will show that this is equivalent to the distribution introduced in DDPM [19] and DDIM [45],[3] up to fixed linear transformations over $\mathbf{x}_t$.

## 3.2 A Diffusion Process for Image Restoration

Similar to SNIPS [25], we consider the singular value decomposition (SVD) of $\boldsymbol{H}$, and perform the diffusion in its spectral space. The idea behind this is to tie the noise present in the measurements $\mathbf{y}$ with the diffusion noise in $\mathbf{x}_{1:T}$, ensuring that the diffusion result $\mathbf{x}_0$ is faithful to the measurements. By using the SVD, we identify the data from $\mathbf{x}$ that is missing in $\mathbf{y}$, and synthesize it using a diffusion process. In conjunction, the noisy data in $\mathbf{y}$ undergoes a denoising process. For example, in inpainting with noise (e.g., $\boldsymbol{H} = \mathrm{diag}([1, \ldots, 1, 0, \ldots, 0])$, $\sigma_{\mathbf{y}} \geq 0$), the spectral space is simply the pixel space, so the model should generate the missing pixels and denoise the observed ones in $\mathbf{y}$. For a general linear $\boldsymbol{H}$, its SVD is given as

$$\boldsymbol{H} = \boldsymbol{U}\boldsymbol{\Sigma}\boldsymbol{V}^\top \tag{3}$$

where $\boldsymbol{U} \in \mathbb{R}^{m \times m}$, $\boldsymbol{V} \in \mathbb{R}^{n \times n}$ are orthogonal matrices, and $\boldsymbol{\Sigma} \in \mathbb{R}^{m \times n}$ is a rectangular diagonal matrix containing the singular values of $\boldsymbol{H}$, ordered descendingly. As this is the case in most useful degradation models, we assume $m \leq n$, but our method would work for $m > n$ as well. We denote the singular values as $s_1 \geq s_2 \geq \ldots \geq s_m$, and define $s_i = 0$ for $i \in [m+1, n]$.

We use the shorthand notations for values in the spectral space: $\bar{\mathbf{x}}_t^{(i)}$ is the $i$-th index of the vector $\bar{\mathbf{x}}_t = \boldsymbol{V}^\top \mathbf{x}_t$, and $\bar{\mathbf{y}}^{(i)}$ is the $i$-th index of the vector $\bar{\mathbf{y}} = \boldsymbol{\Sigma}^\dagger \boldsymbol{U}^\top \mathbf{y}$ (where † denotes the Moore–Penrose pseudo-inverse). Because $\boldsymbol{V}$ is an orthogonal matrix, we can recover $\mathbf{x}_t$ from $\bar{\mathbf{x}}_t$ exactly by left

---

[2]This is called "Variance Exploding" in [47].
[3]This is called "Variance Preserving" in [47].

multiplying $V$. For each index $i$ in $\bar{\mathbf{x}}_t$, we define the variational distribution as:

$$q^{(T)}(\bar{\mathbf{x}}_T^{(i)}|\mathbf{x}_0,\mathbf{y}) = \begin{cases} \mathcal{N}(\bar{\mathbf{y}}^{(i)}, \sigma_T^2 - \frac{\sigma_{\mathbf{y}}^2}{s_i^2}) & \text{if } s_i > 0 \\ \mathcal{N}(\bar{\mathbf{x}}_0^{(i)}, \sigma_T^2) & \text{if } s_i = 0 \end{cases} \tag{4}$$

$$q^{(t)}(\bar{\mathbf{x}}_t^{(i)}|\mathbf{x}_{t+1},\mathbf{x}_0,\mathbf{y}) = \begin{cases} \mathcal{N}(\bar{\mathbf{x}}_0^{(i)} + \sqrt{1-\eta^2}\sigma_t\frac{\bar{\mathbf{x}}_{t+1}^{(i)}-\bar{\mathbf{x}}_0^{(i)}}{\sigma_{t+1}}, \eta^2\sigma_t^2) & \text{if } s_i = 0 \\ \mathcal{N}(\bar{\mathbf{x}}_0^{(i)} + \sqrt{1-\eta^2}\sigma_t\frac{\bar{\mathbf{y}}^{(i)}-\bar{\mathbf{x}}_0^{(i)}}{\sigma_{\mathbf{y}}/s_i}, \eta^2\sigma_t^2) & \text{if } \sigma_t < \frac{\sigma_{\mathbf{y}}}{s_i} \\ \mathcal{N}((1-\eta_b)\bar{\mathbf{x}}_0^{(i)} + \eta_b\bar{\mathbf{y}}^{(i)}, \sigma_t^2 - \frac{\sigma_{\mathbf{y}}^2}{s_i^2}\eta_b^2) & \text{if } \sigma_t \geq \frac{\sigma_{\mathbf{y}}}{s_i} \end{cases} \tag{5}$$

where $\eta \in (0,1]$ is a hyperparameter controlling the variance of the transitions, and $\eta$ and $\eta_b$ may depend on $\sigma_t, s_i, \sigma_{\mathbf{y}}$. We further assume that $\sigma_T \geq \sigma_{\mathbf{y}}/s_i$ for all positive $s_i$.[4]

In the following statement, we show that this construction has the "Gaussian marginals" property similar to the inference distribution used in unconditional diffusion models [19].

**Proposition 3.1.** *The conditional distributions $q^{(t)}$ defined in Equations 4 and 5 satisfy the following:*

$$q(\mathbf{x}_t|\mathbf{x}_0) = \mathcal{N}(\mathbf{x}_0, \sigma_t^2\boldsymbol{I}), \tag{6}$$

*defined by marginalizing over $\mathbf{x}_{t'}$ (for all $t' > t$) and $\mathbf{y}$, where $q(\mathbf{y}|\mathbf{x}_0)$ is defined as in Equation (1) with $\mathbf{x} = \mathbf{x}_0$.*

We place the proof in Appendix C. Intuitively, our construction considers different cases for each index of the spectral space. (*i*) If the corresponding singular value is zero, then $\mathbf{y}$ does not directly provide any information to that index, and the update is similar to regular unconditional generation. (*ii*) If the singular value is non-zero, then the updates consider the information provided by $\mathbf{y}$, which further depends on whether the measurements' noise level in the spectral space ($\sigma_{\mathbf{y}}/s_i$) is larger than the noise level in the diffusion model ($\sigma_t$) or not; the measurements in the spectral space $\bar{\mathbf{y}}^{(i)}$ are then scaled differently for these two cases in order to ensure Proposition 3.1 holds.

Now that we have defined $q^{(t)}$ as a series of Gaussian conditionals, we define our model distribution $p_\theta$ as a series of Gaussian conditionals as well. Similar to DDPM, we aim to obtain predictions of $\mathbf{x}_0$ at every step $t$; and to simplify notations, we use the symbol $\mathbf{x}_{\theta,t}$ to represent this prediction made by a model[5] $f_\theta(\mathbf{x}_{t+1}, t+1) : \mathbb{R}^n \times \mathbb{R} \to \mathbb{R}^n$ that takes in the sample $\mathbf{x}_{t+1}$ and the conditioned time step $(t+1)$. We also define $\bar{\mathbf{x}}_{\theta,t}^{(i)}$ as the $i$-th index of $\bar{\mathbf{x}}_{\theta,t} = \boldsymbol{V}^\top\mathbf{x}_{\theta,t}$.

We define DDRM with trainable parameters $\theta$ as follows:

$$p_\theta^{(T)}(\bar{\mathbf{x}}_T^{(i)}|\mathbf{y}) = \begin{cases} \mathcal{N}(\bar{\mathbf{y}}^{(i)}, \sigma_T^2 - \frac{\sigma_{\mathbf{y}}^2}{s_i^2}) & \text{if } s_i > 0 \\ \mathcal{N}(0, \sigma_T^2) & \text{if } s_i = 0 \end{cases} \tag{7}$$

$$p_\theta^{(t)}(\bar{\mathbf{x}}_t^{(i)}|\mathbf{x}_{t+1},\mathbf{y}) = \begin{cases} \mathcal{N}(\bar{\mathbf{x}}_{\theta,t}^{(i)} + \sqrt{1-\eta^2}\sigma_t\frac{\bar{\mathbf{x}}_{t+1}^{(i)}-\bar{\mathbf{x}}_{\theta,t}^{(i)}}{\sigma_{t+1}}, \eta^2\sigma_t^2) & \text{if } s_i = 0 \\ \mathcal{N}(\bar{\mathbf{x}}_{\theta,t}^{(i)} + \sqrt{1-\eta^2}\sigma_t\frac{\bar{\mathbf{y}}^{(i)}-\bar{\mathbf{x}}_{\theta,t}^{(i)}}{\sigma_{\mathbf{y}}/s_i}, \eta^2\sigma_t^2) & \text{if } \sigma_t < \frac{\sigma_{\mathbf{y}}}{s_i} \\ \mathcal{N}((1-\eta_b)\bar{\mathbf{x}}_{\theta,t}^{(i)} + \eta_b\bar{\mathbf{y}}^{(i)}, \sigma_t^2 - \frac{\sigma_{\mathbf{y}}^2}{s_i^2}\eta_b^2) & \text{if } \sigma_t \geq \frac{\sigma_{\mathbf{y}}}{s_i}. \end{cases} \tag{8}$$

Compared to $q^{(t)}$ in Equations (4) and (5), our definition of $p_\theta^{(t)}$ merely replaces $\bar{\mathbf{x}}_0^{(i)}$ (which we do not know at sampling) with $\bar{\mathbf{x}}_{\theta,t}^{(i)}$ (which depends on our predicted $\mathbf{x}_{\theta,t}$) when $t < T$, and replaces $\bar{\mathbf{x}}_0^{(i)}$ with 0 when $t = T$. It is possible to learn the variances [35] or consider alternative constructions where Proposition 3.1 holds; we leave these options as future work.

### 3.3 "Learning" Image Restoration Models

Once we have defined $p_\theta^{(t)}$ and $q^{(t)}$ by choosing $\sigma_{1:T}$, $\eta$ and $\eta_b$, we can learn model parameters $\theta$ by maximizing the resulting ELBO objective (in Appendix A). However, this approach is not desirable

---

[4]This assumption is fair, as we can set a sufficiently large $\sigma_T$.

[5]Equivalently, the authors of [19] predict the noise values to subtract in order to recover $\mathbf{x}_{\theta,t}$.

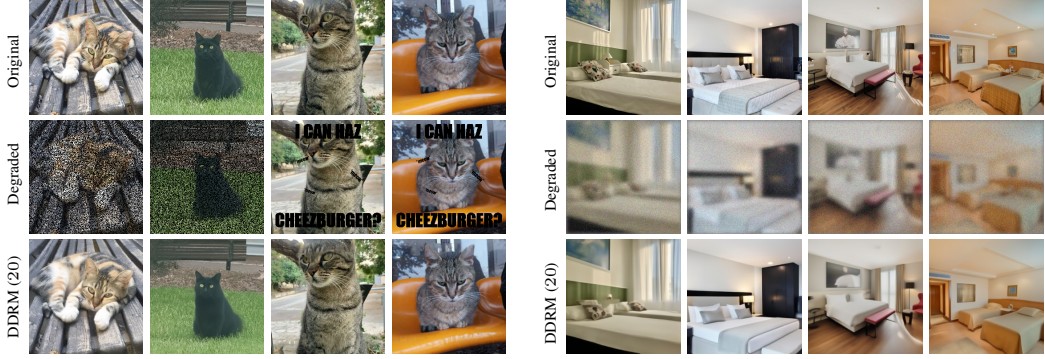

(a) Inpainting results on cat images.    (b) Deblurring results ($\sigma_\mathbf{y} = 0.05$) on bedroom images.

Figure 3: DDRM results on bedroom and cat images, for inpainting and deblurring.

since we have to learn a different model for each inverse problem (given $\mathbf{H}$ and $\sigma_\mathbf{y}$), which is not flexible enough for arbitrary inverse problems. Fortunately, this does not have to be the case. In the following statement, we show that an optimal solution to DDPM / DDIM can also be an optimal solution to a DDRM problem, under reasonable assumptions used in prior work [19, 45].

**Theorem 3.2.** *Assume that the models $f_\theta^{(t)}$ and $f_\theta^{(t')}$ do not have weight sharing whenever $t \neq t'$, then when $\eta = 1$ and $\eta_b = \frac{2\sigma_t^2}{\sigma_t^2 + \sigma_\mathbf{y}^2/s_i^2}$, the ELBO objective of DDRM (details in Appendix A) can be rewritten in the form of the DDPM / DDIM objective in Equation (2).*

We place the proof in Appendix C.

Even for different choices of $\eta$ and $\eta_b$, the proof shows that the DDRM objective is a weighted sum-of-squares error in the spectral space, and thus pre-trained DDPM models are good approximations to the optimal solution. Therefore, we can apply the same diffusion model (*unconditioned* on the inverse problem) using the updates in Equation (7) and Equation (8) and only modify $\mathbf{H}$ and its SVD ($\mathbf{U}, \mathbf{\Sigma}, \mathbf{V}$) for various linear inverse problems.

### 3.4    Accelerated Algorithms for DDRM

Typical diffusion models are trained with many timesteps (*e.g.*, 1000) to achieve optimal unconditional image synthesis quality, but sampling speed is slow as many NFEs are required. Previous works [45, 13] have accelerated this process by "skipping" steps with appropriate update rules. This is also true for DDRM, since we can obtain the denoising autoencoder objective in Equation (2) for any choice of increasing $\sigma_{1:T}$. For a pre-trained diffusion model with $T'$ timesteps, we can choose $\sigma_{1:T}$ to be a subset of the $T'$ steps used in training.

### 3.5    Memory Efficient SVD

Our method, similar to SNIPS [25], utilizes the SVD of the degradation operator $\mathbf{H}$. This constitutes a memory consumption bottleneck in both algorithms as well as other methods such as Plug and Play (PnP) [51], as storing the matrix $\mathbf{V}$ has a space complexity of $\Theta(n^2)$ for signals of size $n$. By leveraging special properties of the matrices $\mathbf{H}$ used, we can reduce this complexity to $\Theta(n)$ for denoising, inpainting, super resolution, deblurring, and colorization (details in Appendix D).

## 4    Related Work

Various deep learning solutions have been suggested for solving inverse problems under different settings (see a detailed survey in [37]). We focus on the *unsupervised* setting, where we have access to a dataset of clean images at training time, but the degradation model is known only at inference time. This setup is inherently general to all linear inverse problems, a property desired in many real-world applications such as medical imaging [46, 20].

Table 1: Noiseless $4\times$ super-resolution and deblurring results on ImageNet 1K ($256 \times 256$).

| Method | $4\times$ super-resolution | | | | Deblurring | | | |
|---|---|---|---|---|---|---|---|---|
| | PSNR↑ | SSIM↑ | KID↓ | NFEs↓ | PSNR↑ | SSIM↑ | KID↓ | NFEs↓ |
| Baseline | 25.65 | 0.71 | 44.90 | **0** | 19.26 | 0.48 | 38.00 | **0** |
| DGP | 23.06 | 0.56 | 21.22 | 1500 | 22.70 | 0.52 | 27.60 | 1500 |
| RED | 26.08 | 0.73 | 53.55 | 100 | 26.16 | 0.76 | 21.21 | 500 |
| SNIPS | 17.58 | 0.22 | 35.17 | 1000 | 34.32 | 0.87 | **0.49** | 1000 |
| DDRM | **26.55** | 0.72 | 7.22 | 20 | 35.64 | 0.95 | 0.71 | 20 |
| DDRM-CC | **26.55** | **0.74** | **6.56** | 20 | **35.65** | **0.96** | 0.70 | 20 |

Almost all unsupervised inverse problem solvers utilize a trained neural network in an iterative scheme. PnP, RED, and their successors [51, 40, 32, 49] apply a denoiser as part of an iterative optimization algorithm such as steepest descent, fixed-point, or alternating direction method of multipliers (ADMM). OneNet [39] trained a network to directly learn the proximal operator of ADMM. A similar use of denoisers in different iterative algorithms is proposed in [34, 16, 30]. The authors of [43] leverages robust classifiers learned with additional class labels.

Another approach is to search the latent space of a generative model for a generated image that, when degraded, is as close as possible to the given measurements. Multiple such methods were suggested, mainly focusing on generative adversarial networks (GANs) [7, 11, 33]. While they exhibit impressive results on images of a specific class, most notably face images, these methods are not shown to be largely successful under a more diverse dataset such as ImageNet [12]. Deep Generative Prior (DGP) mitigates this issue by optimizing the latent input as well as the weights of the GAN's generator [38].

More recently, denoising diffusion models were used to solve inverse problems in both supervised (*i.e.*, degradation model is known during training) [42, 41, 13, 10, 54] and unsupervised settings [22, 26, 25, 21, 46, 47, 9]. Unlike previous approaches, most diffusion-based methods can successfully recover images from measurements with significant noise. However, these methods are very slow, often requiring hundreds or thousands of iterations, and are yet to be proven on diverse datasets. Our method, motivated by variational inference, obtains problem-specific, non-equilibrium update rules that lead to high-quality solutions in much fewer iterations.

ILVR [9] suggests a diffusion-based method that handles noiseless super-resolution, and can run in 250 steps. In Appendix H, we prove that when applied on the same underlying generative diffusion model, ILVR is a special case of DDRM. Therefore, ILVR can be further accelerated to run in 20 steps, but unlike DDRM, it provides no clear way of handling noise in the measurements. Similarly, the authors of [22] suggest a score-based solver for inverse problems that can converge in a small number of iterations, but does not handle noise in the measurements.

## 5 Experiments

### 5.1 Experimental Setup

We demonstrate our algorithm's capabilities using the diffusion models from [19], which are trained on CelebA-HQ [23], LSUN bedrooms, and LSUN cats [56] (all $256 \times 256$ pixels). We test these models on images from FFHQ [24], and pictures from the internet of the considered LSUN category, respectively. In addition, we use the models from [13], trained on the training set of ImageNet $256 \times 256$ and $512 \times 512$, and tested on the corresponding validation set. Some of the ImageNet models require class information. For these models, we use the ground truth labels as input, and denote our algorithm as DDRM class conditional (DDRM-CC). In all experiments, we use $\eta = 0.85$, $\eta_b = 1$, and a uniformly-spaced timestep schedule based on the 1000-step pre-trained models (more details in Appendix E). The number of NFEs (timesteps) is reported in each experiment.

In each of the inverse problems we show, pixel values are in the range $[0, 1]$, and the degraded measurements are obtained as follows: (*i*) for super-resolution, we use a block averaging filter to downscale the images by a factor of 2, 4, or 8 in each axis; (*ii*) for deblurring, the images are blurred

Table 2: $4\times$ super resolution and deblurring results on ImageNet 1K ($256 \times 256$). Input images have an additive noise of $\sigma_{\mathbf{y}} = 0.05$.

| Method | $4\times$ super-resolution | | | | Deblurring | | | |
|---|---|---|---|---|---|---|---|---|
| | PSNR↑ | SSIM↑ | KID↓ | NFEs↓ | PSNR↑ | SSIM↑ | KID↓ | NFEs↓ |
| Baseline | 22.55 | 0.46 | 67.86 | **0** | 18.35 | 0.20 | 75.50 | **0** |
| DGP | 20.69 | 0.43 | 42.17 | 1500 | 21.20 | 0.45 | 34.02 | 1500 |
| RED | 22.90 | 0.49 | 43.45 | 100 | 14.69 | 0.08 | 121.82 | 500 |
| SNIPS | 16.30 | 0.14 | 67.77 | 1000 | 16.37 | 0.14 | 77.96 | 1000 |
| DDRM | 25.21 | 0.66 | 12.43 | 20 | 25.45 | 0.66 | 15.24 | 20 |
| DDRM-CC | **25.22** | **0.67** | **10.82** | 20 | **25.46** | **0.67** | **13.49** | 20 |

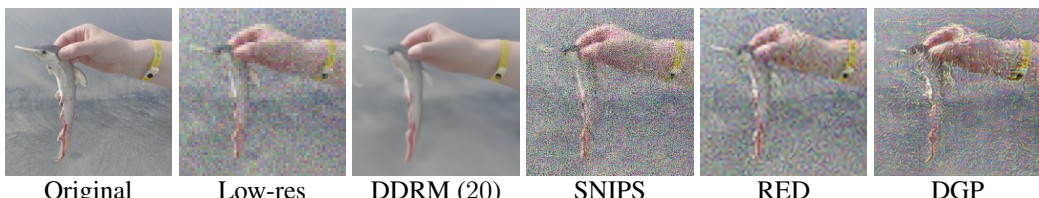

| Original | Low-res | DDRM (20) | SNIPS | RED | DGP |
|---|---|---|---|---|---|

Figure 4: $4\times$ noisy super resolution comparison with $\sigma_{\mathbf{y}} = 0.05$.

by a $9 \times 9$ uniform kernel, and singular values below a certain threshold are zeroed, making the problem more ill-posed. (*iii*) for colorization, the grayscale image is an average of the red, green, and blue channels of the original image; (*iv*) and for inpainting, we mask parts of the original image with text overlay or randomly drop $50\%$ of the pixels. Additive white Gaussian noise can optionally be added to the measurements in all inverse problems. We additionally conduct experiments on bicubic super-resolution and deblurring with an anisotropic Gaussian kernel in Appendix I.

Our code is available at `https://github.com/bahjat-kawar/ddrm`.

## 5.2 Quantitative Experiments

In order to quantify DDRM's performance, we focus on the ImageNet dataset ($256 \times 256$) for its diversity. For each experiment, we report the average peak signal-to-noise ratio (PSNR) and structural similarity index measure (SSIM) [52] to measure faithfulness to the original image, and the kernel Inception distance (KID) [5], multiplied by $10^3$, to measure the resulting image quality.

We compare DDRM (with 20 and 100 steps) with other unsupervised methods that work in reasonable time (requiring 1500 NFEs or less) and can operate on ImageNet. Namely, we compare with RED

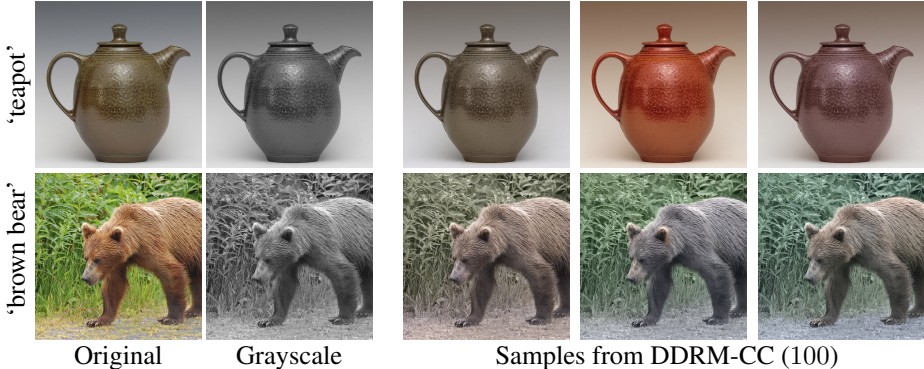

'teapot'
'brown bear'

| Original | Grayscale | Samples from DDRM-CC (100) | | |
|---|---|---|---|---|

Figure 5: $512 \times 512$ ImageNet colorization. DDRM-CC produces various samples for multiple runs on the same input.

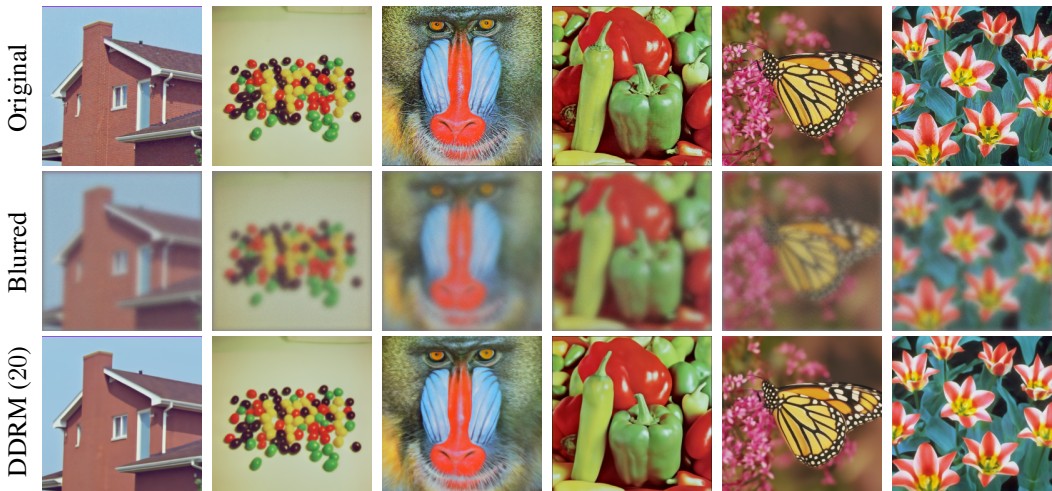

Figure 6: Results on $256 \times 256$ USC-SIPI images using an ImageNet model. Blurred images have a noise of $\sigma_{\mathbf{y}} = 0.01$.

[40], DGP [38], and SNIPS [25]. The exact setup of each method is detailed in Appendix F. We used the same hyperparameters for noisy and noiseless versions of the same problem for DGP, RED, and SNIPS, as tuning them for each version would compromise their unsupervised nature. Nevertheless, the performance of baselines like RED with such a tuning does not surpass that of DDRM, as we show in Appendix F. In addition, we show upscaling by bicubic interpolation as a baseline for super-resolution, and the blurry image itself as a baseline for deblurring. OneNet [39] is not included in the comparisons as it is limited to images of size $64 \times 64$, and generalization to higher dimensions requires an improved network architecture.

We evaluate all methods on the problems of $4\times$ super-resolution and deblurring, on one validation set image from each of the 1000 ImageNet classes, following [38]. Table 1 shows that DDRM outperforms all baseline methods, in all metrics, and on both problems with only 20 steps. The only exception to this is that SNIPS achieves better KID than DDRM in noiseless deblurring, but it requires $50\times$ more NFEs to do so. Note that the runtime of all the tested methods is perfectly linear with NFEs, with negligible differences in time per iteration. DGP and DDRM-CC use ground-truth class labels for the test images to aid in the restoration process, and thus have an unfair advantage.

DDRM's appeal compared to previous methods becomes more substantial when significant noise is added to the measurements. Under this setting, DGP, RED, and SNIPS all fail to produce viable results, as evident in Table 2 and Figure 4. Since DDRM is fast, we also evaluate it on the entire ImageNet validation set in Appendix F.

### 5.3 Qualitative Experiments

DDRM produces high quality reconstructions across all the tested datasets and problems, as can be seen in Figures 1 and 3, and in Appendix I. As it is a posterior sampling algorithm, DDRM can produce multiple outputs for the same input, as demonstrated in Figure 5. Moreover, the unconditional ImageNet diffusion models can be used to solve inverse problems on out-of-distribution images with general content. In Figure 6, we show DDRM successfully restoring $256 \times 256$ images from USC-SIPI [53] that do not necessarily belong to any ImageNet class (more results in Appendix I).

## 6 Conclusions

We have introduced DDRM, a general sampling-based linear inverse problem solver based on unconditional/class-conditional diffusion generative models as learned priors. Motivated by variational inference, DDRM only requires a few number of NFEs (*e.g.*, 20) compared to other sampling-based baselines (*e.g.*, 1000 for SNIPS) and achieves scalability in multiple useful scenarios, including denoising, super-resolution, deblurring, inpainting, and colorization. We demonstrate the empirical

successes of DDRM on various problems and datasets, including general natural images outside the distribution of the observed training set. To our best knowledge, DDRM is the first unsupervised method that effectively and efficiently samples from the posterior distribution of inverse problems with significant noise, and can work on natural images with general content.

In terms of future work, apart from further optimizing the timestep and variance schedules, it would be interesting to investigate the following: (*i*) applying DDRM to non-linear inverse problems, (*ii*) addressing scenarios where the degradation operator is unknown, and (*iii*) self-supervised training techniques inspired by DDRM as well as ones used in supervised techniques [41] that further improve performance of unsupervised models for image restoration.

## Acknowledgements

We thank Kristy Choi, Charlie Marx, and Avital Shafran for insightful discussions and feedback. This research was supported by NSF (#1651565, #1522054, #1733686), ONR (N00014-19-1-2145), AFOSR (FA9550-19-1-0024), ARO (W911NF-21-1-0125), Sloan Fellowship, Amazon AWS, Stanford Institute for Human-Centered Artificial Intelligence (HAI), Google Cloud, the Israel Science Foundation (ISF) under Grant 335/18, the Israeli Council For Higher Education - Planning & Budgeting Committee, and the Stephen A. Kreynes Fellowship.

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
