# A  Details of the DDRM ELBO objective

DDRM is a Markov chain conditioned on $\mathbf{y}$, which would lead to the following ELBO objective [45]:

$$\mathbb{E}_{\mathbf{x}_0 \sim q(\mathbf{x}_0), \mathbf{y} \sim q(\mathbf{y}|\mathbf{x}_0)}[\log p_\theta(\mathbf{x}_0|\mathbf{y})] \tag{9}$$

$$\geq -\mathbb{E}\left[\sum_{t=1}^{T-1} D_{\mathrm{KL}}(q^{(t)}(\mathbf{x}_t|\mathbf{x}_{t+1}, \mathbf{x}_0, \mathbf{y})\|p_\theta^{(t)}(\mathbf{x}_t|\mathbf{x}_{t+1}, \mathbf{y}))\right] + \mathbb{E}\left[\log p_\theta^{(0)}(\mathbf{x}_0|\mathbf{x}_1, \mathbf{y})\right]$$

$$- \mathbb{E}[D_{\mathrm{KL}}(q^{(T)}(\mathbf{x}_T|\mathbf{x}_0, \mathbf{y})\|p_\theta^{(T)}(\mathbf{x}_T|\mathbf{y}))] \tag{10}$$

where $q(\mathbf{x}_0)$ is the data distribution, $q(\mathbf{y}|\mathbf{x}_0)$ follows Equation 1 in the main paper, the expectation on the right hand side is given by sampling $\mathbf{x}_0 \sim q(\mathbf{x}_0)$, $\mathbf{y} \sim q(\mathbf{y}|\mathbf{x}_0)$, $\mathbf{x}_T \sim q^{(T)}(\mathbf{x}_T|\mathbf{x}_0, \mathbf{y})$, and $\mathbf{x}_t \sim q^{(t)}(\mathbf{x}_t|\mathbf{x}_{t+1}, \mathbf{x}_0, \mathbf{y})$ for $t \in [1, T-1]$.

# B  Equivalence between "Variance Preserving" and "Variance Exploding" Diffusion Models

In our main paper, we describe our methods based on the "Variance Exploding" hyperparameters $\sigma_t$, where $\sigma_t \in [0, \infty)$ and

$$q(\mathbf{x}_t|\mathbf{x}_0) = \mathcal{N}(\mathbf{x}_0, \sigma_t^2 \boldsymbol{I}). \tag{11}$$

In DDIM [45], the hyperparameters are "Variance Preserving" ones $\alpha_t$, where $\alpha_t \in (0, 1]$ and

$$q(\mathbf{x}_t|\mathbf{x}_0) = \mathcal{N}(\sqrt{\alpha_t}\mathbf{x}_0, (1 - \alpha_t)\boldsymbol{I}). \tag{12}$$

We use the colored notation $\mathbf{x}_t$ to emphasize that this is different from $\mathbf{x}_t$ (an exception is $\mathbf{x}_0 = \mathbf{x}_0$). Using the reparametrization trick, we have that:

$$\mathbf{x}_t = \mathbf{x}_0 + \sigma_t \epsilon \tag{13}$$

$$\mathbf{x}_t = \sqrt{\alpha_t}\mathbf{x}_0 + \sqrt{1 - \alpha_t}\epsilon \tag{14}$$

where $\epsilon \sim \mathcal{N}(0, \boldsymbol{I})$. We can divide by $\sqrt{1 + \sigma_t^2}$ in both sides of Equation 13:

$$\frac{\mathbf{x}_t}{\sqrt{1 + \sigma_t^2}} = \frac{\mathbf{x}_0}{\sqrt{1 + \sigma_t^2}} + \frac{\sigma_t}{\sqrt{1 + \sigma_t^2}}\epsilon. \tag{15}$$

Let $\alpha_t = 1/(1 + \sigma_t^2)$, and let $\mathbf{x}_t = \mathbf{x}_t/\sqrt{1 + \sigma_t^2}$; then from Equation 15 we have that

$$\mathbf{x}_t = \sqrt{\alpha_t}\mathbf{x}_0 + \sqrt{1 - \alpha_t}\epsilon, \tag{16}$$

which is equivalent to the "Variance Preserving" case. Therefore, we can use "Variance Preserving" models, such as DDPM, directly in our DDRM updates, even though the latter uses the "Variance Exploding" parametrization:

1. From $\mathbf{x}_t$, obtain predictions $\epsilon$ and $\mathbf{x}_t = \mathbf{x}_t\sqrt{1 + \sigma_t^2}$.
2. From $\mathbf{x}_t$ and $\epsilon$, apply DDRM updates to get $\mathbf{x}_{t-1}$.
3. From $\mathbf{x}_{t-1}$, get $\mathbf{x}_{t-1} = \mathbf{x}_{t-1}/\sqrt{1 + \sigma_{t-1}^2}$.

Note that although the inference algorithms are shown to be equivalent, the choice between "Variance Preserving" and "Variance Exploding" may affect the training of diffusion networks.

# C  Proofs

**Proposition 3.1.** *The conditional distributions $q^{(t)}$ defined in Equations 4 and 5 satisfy the following:*

$$q(\mathbf{x}_t|\mathbf{x}_0) = \mathcal{N}(\mathbf{x}_0, \sigma_t^2 \boldsymbol{I}), \tag{6}$$

*defined by marginalizing over $\mathbf{x}_{t'}$ (for all $t' > t$) and $\mathbf{y}$, where $q(\mathbf{y}|\mathbf{x}_0)$ is defined as in Equation (1) with $\mathbf{x} = \mathbf{x}_0$.*

*Proof.* The proof uses a basic property of Gaussian marginals (see [4] for the complete version).

1. If $p(z_1|z_0) = \mathcal{N}(z_0, V_1)$, $p(z_2|z_1) = \mathcal{N}(\alpha z_1, V_2)$, then $p(z_2|z_0) = \mathcal{N}(\alpha z_0, \alpha^2 V_1 + V_2)$.

2. If $p(z_1) = \mathcal{N}(\mu_1, V_1)$ and $p(z_2) = \mathcal{N}(\mu_2, V_2)$, then $p(z_1 + z_2) = \mathcal{N}(\mu_1 + \mu_2, V_1 + V_2)$.

First, we note that $q(\mathbf{y}|\mathbf{x}_0)$ is defined from Equation 1 in the main paper, and thus for all $i$:

$$q(\bar{\mathbf{y}}^{(i)}|\mathbf{x}_0) = \mathcal{N}(\bar{\mathbf{x}}_0^{(i)}, \sigma_{\mathbf{y}}^2/s_i^2). \tag{17}$$

**Case I** For $\mathbf{x}_T$, it is obvious when $s_i = 0$. When $s_i > 0$, we have Equation 17 and that:

$$q^{(T)}(\bar{\mathbf{x}}_T^{(i)}|\mathbf{x}_0, \mathbf{y}) = \mathcal{N}(\bar{\mathbf{y}}^{(i)}, \sigma_T^2 - \frac{\sigma_{\mathbf{y}}^2}{s_i^2}), \tag{18}$$

and thus

$$q^{(T)}(\bar{\mathbf{x}}_T^{(i)}|\mathbf{x}_0) = \mathcal{N}(\bar{\mathbf{x}}_0^{(i)}, \sigma_{\mathbf{y}}^2/s_i^2 + \sigma_T^2 - \frac{\sigma_{\mathbf{y}}^2}{s_i^2}) = \mathcal{N}(\bar{\mathbf{x}}_0^{(i)}, \sigma_T^2).$$

**Case II** For any $t < T$ and $i$ such that $s_i > 0$ and $\sigma_t > \sigma_{\mathbf{y}}/s_i$, we have Equation 17 and that:

$$q^{(t)}(\bar{\mathbf{x}}_t^{(i)}|\mathbf{x}_{t+1}, \mathbf{x}_0, \mathbf{y}) = \mathcal{N}\left((1 - \eta_b)\bar{\mathbf{x}}_0^{(i)} + \eta_b\bar{\mathbf{y}}^{(i)}, \sigma_t^2 - \frac{\sigma_{\mathbf{y}}^2}{s_i^2}\eta_b^2\right), \tag{19}$$

and thus we can safely remove the dependence on $\mathbf{x}_{t+1}$ via marginalization. $q^{(t)}(\bar{\mathbf{x}}_t^{(i)}|\mathbf{x}_0)$ is a Gaussian with the mean being $(1 - \eta_b)\bar{\mathbf{x}}_0^{(i)} + \eta_b\bar{\mathbf{x}}_0^{(i)} = \bar{\mathbf{x}}_0^{(i)}$ and variance being

$$\sigma_t^2 - \frac{\sigma_{\mathbf{y}}^2}{s_i^2}\eta_b^2 + \frac{\sigma_{\mathbf{y}}^2}{s_i^2}\eta_b^2 = \sigma_t^2,$$

where we note that $\bar{\mathbf{y}}^{(i)}$ has a standard deviation of $\sigma_{\mathbf{y}}/s_i$.

**Case III** For any $t < T$ and $i$ such that $s_i > 0$ and $\sigma_t < \sigma_{\mathbf{y}}/s_i$, we have Equation 17, so $(\bar{\mathbf{y}}^{(i)} - \bar{\mathbf{x}}_0^{(i)})/(\sigma_{\mathbf{y}}/s_i)$ is distributed as a standard Gaussian. Moreover, similar to **Case II**, $q^{(t)}(\bar{\mathbf{x}}_t^{(i)}|\mathbf{x}_0)$ is a Gaussian with its mean being

$$\bar{\mathbf{x}}_0^{(i)} + \sqrt{1 - \eta^2}\sigma_t\frac{\bar{\mathbf{y}}^{(i)} - \bar{\mathbf{x}}_0^{(i)}}{\sigma_{\mathbf{y}}/s_i}$$

and its variance being $\eta^2\sigma_t^2$, so $q^{(t)}(\bar{\mathbf{x}}_t^{(i)}|\mathbf{x}_0)$ is a Gaussian with a mean of $\bar{\mathbf{x}}_0^{(i)}$ and a variance of

$$(1 - \eta^2)\sigma_t^2 + \eta^2\sigma_t^2 = \sigma_t^2.$$

**Case IV** For any $t \leq T$ and $i$ such that $s_i = 0$ (where there is no dependence on $\mathbf{y}$), we apply mathematical induction. The base case ($t = T$) is true, as we have shown earlier in **Case I**. In the step case ($t < T$), we have that $q^{(t+1)}(\bar{\mathbf{x}}_{t+1}^{(i)}|\mathbf{x}_0) = \mathcal{N}(\bar{\mathbf{x}}_0^{(i)}, \sigma_{t+1}^2)$. Similar to **Case II**, $q^{(t)}(\bar{\mathbf{x}}_t^{(i)}|\mathbf{x}_0)$ is a Gaussian with its mean being

$$\bar{\mathbf{x}}_0^{(i)} + \sqrt{1 - \eta^2}\sigma_t\frac{\bar{\mathbf{x}}_{t+1}^{(i)} - \bar{\mathbf{x}}_0^{(i)}}{\sigma_{t+1}}$$

and variance being $\eta^2\sigma_t^2$, which does not depend on $\mathbf{y}$. Therefore, $q^{(t)}(\bar{\mathbf{x}}_t^{(i)}|\mathbf{x}_0)$ is also Gaussian, with a mean of $\bar{\mathbf{x}}_0^{(i)}$ and a variance of

$$(1 - \eta^2)\sigma_t^2 + \eta^2\sigma_t^2 = \sigma_t^2.$$

Hence, the proof is completed via the four cases. □

**Theorem 3.2.** *Assume that the models $f_\theta^{(t)}$ and $f_\theta^{(t')}$ do not have weight sharing whenever $t \neq t'$, then when $\eta = 1$ and $\eta_b = \frac{2\sigma_t^2}{\sigma_t^2 + \sigma_{\mathbf{y}}^2/s_i^2}$, the ELBO objective of DDRM (details in Appendix A) can be rewritten in the form of the DDPM / DDIM objective in Equation (2).*

*Proof.* As there is no parameter sharing between models at different time steps $t$, let us focus on any particular time step $t$ and rewrite the corresponding objective as a denoising autoencoder objective.

**Case I** For $t > 0$, the only term in Equation 10 that is related to $f_\theta^{(t)}$ (which is used to make the prediction $\mathbf{x}_{\theta,t}$) is:

$$
\begin{aligned}
&D_{\mathrm{KL}}(q^{(t)}(\mathbf{x}_t|\mathbf{x}_{t+1},\mathbf{x}_0,\mathbf{y})\|p_\theta^{(t)}(\mathbf{x}_t|\mathbf{x}_{t+1},\mathbf{y})) \\
=\ &D_{\mathrm{KL}}(q^{(t)}(\bar{\mathbf{x}}_t|\mathbf{x}_{t+1},\mathbf{x}_0,\mathbf{y})\|p_\theta^{(t)}(\bar{\mathbf{x}}_t|\mathbf{x}_{t+1},\mathbf{y})) \\
=\ &\sum_{i=1}^n D_{\mathrm{KL}}(q^{(t)}(\bar{\mathbf{x}}_t^{(i)}|\mathbf{x}_{t+1},\mathbf{x}_0,\mathbf{y})\|p_\theta^{(t)}(\bar{\mathbf{x}}_t^{(i)}|\mathbf{x}_{t+1},\mathbf{x}_0,\mathbf{y})),
\end{aligned}
\tag{20}
$$

where the first equality is from the orthogonality of $V^\top$ and the second equality is from the fact that both $q^{(t)}$ and $p_\theta^{(t)}$ over the spectral space are Gaussians with identical diagonal covariance matrices (so the KL divergence can factorize).

Here, we will use a simple property of the KL divergence between univariate Gaussians [27]:

If $p = \mathcal{N}(\mu_1, V_1)$, $q = \mathcal{N}(\mu_2, V_2)$, then

$$
D_{\mathrm{KL}}(p\|q) = \frac{1}{2}\log\frac{V_2}{V_1} + \frac{V_1 + (\mu_1 - \mu_2)^2}{2V_2} - \frac{1}{2}.
$$

Since we constructed $p_\theta^{(t)}$ and $q^{(t)}$ to have the same variance, Equation 20 is a total squared error with weights for each dimension of $\bar{\mathbf{x}}_t$ (the spectral space), so the DDPM objective (which is a total squared error objective in the original space) is still a good approximation. In order to transform it into a denoising autoencoder objective (equivalent to DDPM), the weights have to be equal. Next, we will show that our construction of $\eta = 1$ and $\eta_b = 2\sigma_t^2/(\sigma_t^2 + \sigma_\mathbf{y}^2/s_i^2)$ satisfies this.

All the indices $i$ will fall into one of the three cases: $s_i = 0$, $\sigma_t < \sigma_\mathbf{y}/s_i$, or $\sigma_t > \sigma_\mathbf{y}/s_i$.

- For $s_i = 0$, the KL divergence is $\frac{(\bar{\mathbf{x}}_{\theta,t}^{(i)} - \bar{\mathbf{x}}_0^{(i)})^2}{2\sigma_t^2}$, where we recall $\bar{\mathbf{x}}_{\theta,t} = V^\top f_\theta^{(t)}(\mathbf{x}_{t+1})$.

- For $\sigma_t < \frac{\sigma_\mathbf{y}}{s_i}$, the KL divergence is also $\frac{(\bar{\mathbf{x}}_{\theta,t}^{(i)} - \bar{\mathbf{x}}_0^{(i)})^2}{2\sigma_t^2}$.

- For $\sigma_t \geq \frac{\sigma_\mathbf{y}}{s_i}$, we have defined $\eta_b$ as a solution to the following quadratic equation (the other solution is $0$, which is irrelevant to our case since it does not make use of information from $\mathbf{y}$):

$$
(\sigma_t^2 + \frac{\sigma_\mathbf{y}^2}{s_i^2})\eta_b^2 - 2\sigma_t^2\eta_b = 0;
\tag{21}
$$

reorganizing terms, we have that:

$$
(\sigma_t^2 + \frac{\sigma_\mathbf{y}^2}{s_i^2})\eta_b^2 - 2\sigma_t^2\eta_b + \sigma_t^2 = \sigma_t^2
$$

$$
\sigma_t^2(1 - \eta_b)^2 = \sigma_t^2\eta_b^2 - 2\sigma_t^2\eta_b + \sigma_t^2 = \sigma_t^2 - \frac{\sigma_\mathbf{y}^2}{s_i^2}\eta_b^2
$$

$$
\frac{(1 - \eta_b)^2}{\sigma_t^2 - \frac{\sigma_\mathbf{y}^2}{s_i^2}\eta_b^2} = \frac{1}{\sigma_t^2},
\tag{22}
$$

So the KL divergence is

$$
\frac{(1 - \eta_b)^2}{2(\sigma_t^2 - \frac{\sigma_\mathbf{y}^2}{s_i^2}\eta_b^2)}(\bar{\mathbf{x}}_{\theta,t}^{(i)} - \bar{\mathbf{x}}_0^{(i)})^2 = \frac{(\bar{\mathbf{x}}_{\theta,t}^{(i)} - \bar{\mathbf{x}}_0^{(i)})^2}{2\sigma_t^2}.
$$

Therefore, regardless of how the cases are distributed among indices, we will always have that:

$$D_{\mathrm{KL}}(q^{(t)}(\bar{\mathbf{x}}_t|\mathbf{x}_{t+1}, \mathbf{x}_0, \mathbf{y}) \| p_\theta^{(t)}(\bar{\mathbf{x}}_t|\mathbf{x}_{t+1}, \mathbf{y})) = \sum_{i=1}^{n^2} \frac{(\bar{\mathbf{x}}_{\theta,t}^{(i)} - \bar{\mathbf{x}}_0^{(i)})^2}{2\sigma_t^2} = \frac{\|\bar{\mathbf{x}}_{\theta,t} - \bar{\mathbf{x}}_0\|_2^2}{2\sigma_t^2} = \frac{\|f_\theta^{(t)}(\mathbf{x}_{t+1}) - \mathbf{x}_0\|_2^2}{2\sigma_t^2}.$$

**Case II** For $t = 0$, we will only have two cases ($s_i = 0$ or $\sigma_t < \frac{\sigma_\mathbf{y}}{s_i}$), and thus, similar to **Case I**,

$$\log p_\theta^{(0)}(\bar{\mathbf{x}}_0|\mathbf{x}_1, \mathbf{y}) = \sum_{i=1}^{n^2} \log p_\theta^{(0)}(\bar{\mathbf{x}}_0^{(i)}|\mathbf{x}_1, \mathbf{y}) \propto \sum_{i=1}^{n^2}(\bar{\mathbf{x}}_{\theta,0}^{(i)} - \bar{\mathbf{x}}_0^{(i)})^2 = \|\bar{\mathbf{x}}_{\theta,0} - \bar{\mathbf{x}}_0\|_2^2 = \|f_\theta^{(0)}(\mathbf{x}_1) - \mathbf{x}_0\|_2^2,$$

as long as we have a constant variance for $p_\theta^{(0)}$. Thus, every individual term in Equation 10 can be written as a denoising autoencoder objective, completing the proof. $\qquad\square$

# D    Memory Efficient SVD

Here we explain how we obtained the singular value decomposition (SVD) for different degradation models efficiently.

## D.1    Denoising

In denoising, the corrupted image is the original image with additive white Gaussian noise. Therefore, $\boldsymbol{H} = \boldsymbol{I}$ and all the SVD elements of $\boldsymbol{H}$ are simply the identity matrix $\boldsymbol{I}$, which in turns makes their multiplication by different vectors trivial.

## D.2    Inpainting

In inpainting, $\boldsymbol{H}$ retains a known subset of size $k$ of the image's pixels. This is equivalent to permuting the pixels such that the retained one are placed at the top, then keeping the first $k$ entries. Therefore,

$$\boldsymbol{H} = \boldsymbol{I}\boldsymbol{\Sigma}\boldsymbol{P}, \tag{23}$$

where $\boldsymbol{P}$ is the appropriate permutation matrix, $\boldsymbol{\Sigma}$ is a rectangular diagonal matrix of size $k \times n$ with ones in its main diagonal, and $\boldsymbol{I}$ is the identity matrix. Since permutation matrices are orthogonal, Equation 23 is the SVD of $\boldsymbol{H}$.

We can multiply a given vector by $\boldsymbol{P}$ and $\boldsymbol{P}^T$ by storing the permutation itself rather than the matrix. $\boldsymbol{\Sigma}$ can multiply a vector by simply slicing it. Therefore, by storing the appropriate permutation and the number $k$, we can apply each element of the SVD with $\Theta(n)$ space complexity.

## D.3    Super Resolution

For super resolution, we assume that the original image of size $d \times d$ (*i.e.* $n = 3d^2$) is downscaled using a block averaging filter by $r$ in each dimension, such that $d$ is divisible by $r$. In this scenario, each pixel in the output image is the average of an $r \times r$ patch in the input image, and each such patch affects exactly one output pixel. Therefore, any output pixel is given by $(\boldsymbol{H}\mathbf{x})_i = \boldsymbol{k}^T\boldsymbol{p}_i$, where $\boldsymbol{k}$ is a vector of size $r^2$ with $\frac{1}{r^2}$ in each entry, and $\boldsymbol{p}_i$ is the vectorized $i$-th $r \times r$ patch. More formally, if $\boldsymbol{P}_1$ is a permutation matrix that reorders a vectorized image into patches, then

$$\boldsymbol{H} = (\boldsymbol{I} \otimes \boldsymbol{k}^T)\, \boldsymbol{P}_1,$$

where $\otimes$ is the Kronecker product, and $\boldsymbol{I}$ is the identity matrix of size $\frac{d}{r} \times \frac{d}{r}$. In order to obtain the SVD of $\boldsymbol{H}$, we calculate the SVD of $\boldsymbol{k}^T$:

$$\boldsymbol{k}^T = \boldsymbol{U_k}\boldsymbol{\Sigma_k}\boldsymbol{V_k}^T.$$

Using properties of the Kronecker product, we observe

$$\boldsymbol{H} = (\boldsymbol{I} \otimes \boldsymbol{k}^T)\, \boldsymbol{P}_1 = ((\boldsymbol{I}\boldsymbol{I}\boldsymbol{I}) \otimes (\boldsymbol{U_k}\boldsymbol{\Sigma_k}\boldsymbol{V_k}^T))\, \boldsymbol{P}_1 \tag{24}$$
$$= (\boldsymbol{I} \otimes \boldsymbol{U_k})\, (\boldsymbol{I} \otimes \boldsymbol{\Sigma_k})\, (\boldsymbol{I} \otimes \boldsymbol{V_k}^T)\, \boldsymbol{P}_1.$$

The Kronecker product of two orthogonal matrices is an orthogonal matrix. Therefore, $\boldsymbol{I} \otimes \boldsymbol{U_k}$ and $\boldsymbol{I} \otimes \boldsymbol{V_k}^T$ are orthogonal. Observe that the matrix $\boldsymbol{I} \otimes \boldsymbol{\Sigma_k}$ has one non-zero value ($\frac{1}{r^2}$) in each row. By applying a simple permutation on its columns, these values can be reordered to be on the main diagonal. We denote the appropriate permutation matrix by $\boldsymbol{P}_2$, and obtain

$$\boldsymbol{H} = \boldsymbol{U\Sigma V}^T, \tag{25}$$

where $\boldsymbol{U} = \boldsymbol{I} \otimes \boldsymbol{U_k}$ is orthogonal, $\boldsymbol{\Sigma} = (\boldsymbol{I} \otimes \boldsymbol{\Sigma_k}) \boldsymbol{P}_2^T$ is a rectangular diagonal matrix with non-negative entries, and $\boldsymbol{V}^T = \boldsymbol{P}_2 \left( \boldsymbol{I} \otimes \boldsymbol{V_k}^T \right) \boldsymbol{P}_1$ is orthogonal. As such, Equation 25 is the SVD of $\boldsymbol{H}$. By storing the permutations and the SVD elements of $\boldsymbol{k}^T$, we can simulate each element of the SVD of $\boldsymbol{H}$ with $\Theta(n)$ space complexity, without directly calculating the Kronecker products with $\boldsymbol{I}$.

### D.4   Colorization

The grayscale image is obtained by averaging the red, green, and blue channels of each pixel. This means that every output pixel is given by $(\boldsymbol{Hx})_i = \boldsymbol{k}^T \boldsymbol{p}_i$, where $\boldsymbol{k}^T = \left( \frac{1}{3} \quad \frac{1}{3} \quad \frac{1}{3} \right)$ and $\boldsymbol{p}_i$ is the 3-valued $i$-th pixel of the original color image. The SVD of $\boldsymbol{H}$ is obtained exactly the same as in the super resolution case, with separate pixels replacing separate patches.

### D.5   Deblurring

We focus on *separable blurring*, where the 2D blurring kernel is $\boldsymbol{K} = \boldsymbol{r}\boldsymbol{c}^T$, which means $\boldsymbol{c}$ is applied on the columns of the image, and $\boldsymbol{r}^T$ is applied on its rows. The blurred image can be obtained by $\boldsymbol{B} = \boldsymbol{A}_c \boldsymbol{X} \boldsymbol{A}_r^T$, where $\boldsymbol{A}_c$ and $\boldsymbol{A}_r$ apply a 1D convolution with kernels $\boldsymbol{c}$ and $\boldsymbol{r}$, respectively. Alternatively, $\boldsymbol{b} = \boldsymbol{Hx}$, where $\boldsymbol{x}$ is the vectorized image $\boldsymbol{X}$, $\boldsymbol{b}$ is the vectorized blurred image $\boldsymbol{B}$, and $\boldsymbol{H}$ is the matrix applying the 2D convolution $\boldsymbol{K}$. It can be shown that $\boldsymbol{H} = \boldsymbol{A}_r \otimes \boldsymbol{A}_c$, where $\otimes$ is the Kronecker product. In order to calculate the SVD of $\boldsymbol{H}$, we calculate the SVD of $\boldsymbol{A}_r$ and $\boldsymbol{A}_c$:

$$\boldsymbol{A}_r = \boldsymbol{U}_r \boldsymbol{\Sigma}_r \boldsymbol{V}_r^T, \quad \boldsymbol{A}_c = \boldsymbol{U}_c \boldsymbol{\Sigma}_c \boldsymbol{V}_c^T.$$

Using the properties of the Kronecker product, we observe

$$\begin{aligned}
\boldsymbol{H} = \boldsymbol{A}_r \otimes \boldsymbol{A}_c &= \left( \boldsymbol{U}_r \boldsymbol{\Sigma}_r \boldsymbol{V}_r^T \right) \otimes \left( \boldsymbol{U}_c \boldsymbol{\Sigma}_c \boldsymbol{V}_c^T \right) \\
&= \left( \boldsymbol{U}_r \otimes \boldsymbol{U}_c \right) \left( \boldsymbol{\Sigma}_r \otimes \boldsymbol{\Sigma}_c \right) \left( \boldsymbol{V}_r \otimes \boldsymbol{V}_c \right)^T .
\end{aligned} \tag{26}$$

The Kronecker product preserves orthogonality. Therefore, Equation 26 is a valid SVD of $\boldsymbol{H}$, with the exception of the singular values not being on the main diagonal, and not being sorted descendingly. We reorder the columns so that the singular values are on the main diagonal and denote the corresponding permutation matrix by $\boldsymbol{P}_1$. We also sort the values descendingly and denote the sorting permutation matrix by $\boldsymbol{P}_2$, and obtain the following SVD:

$$\boldsymbol{H} = \boldsymbol{U\Sigma V}^T, \tag{27}$$

where $\boldsymbol{U} = \left( \boldsymbol{U}_r \otimes \boldsymbol{U}_c \right) \boldsymbol{P}_2^T$, $\boldsymbol{\Sigma} = \boldsymbol{P}_2 \left( \boldsymbol{\Sigma}_r \otimes \boldsymbol{\Sigma}_c \right) \boldsymbol{P}_1^T \boldsymbol{P}_2^T$, and $\boldsymbol{V}^T = \boldsymbol{P}_2 \boldsymbol{P}_1 \left( \boldsymbol{V}_r \otimes \boldsymbol{V}_c \right)^T$.

For every matrix of the form $\boldsymbol{M} = \boldsymbol{N} \otimes \boldsymbol{L}$, it holds that $\boldsymbol{Mx}$ is the vectorized version of $\boldsymbol{LXN}^T$. By using this property and applying the relevant permutation, we can simulate multiplying a vector by $\boldsymbol{U}, \boldsymbol{V}, \boldsymbol{U}^T$, or $\boldsymbol{V}^T$ without storing the full matrix. The space complexity of this approach is $\Theta(n)$, which is required for computing the SVD of $\boldsymbol{A}_r$ and $\boldsymbol{A}_c$, as well as storing the permutations.

The above calculations remain valid when the blurring is zero-padded, *i.e.*, images are padded with zeroes so that the convolution is not circulant around the edges. We consider a zero-padded deblurring problem in our experiments. Note that the noiseless version of this problem has a simple solution – applying the pseudo-inverse of the blurring matrix on the blurry image. This solution attains 32.41dB in PSNR on ImageNet-1K, while DDRM improves upon it and achieves 35.64dB. When noise is added to the blurry image, such a simple solution amplifies the noise and fails to provide a valid output. Therefore, we opt not to report its results.

Furthermore, the above calculations are also applicable to blurring with strided convolutions. We use this fact in our implementation of the bicubic super resolution SVD, which can be interpreted as a strided convolution with a fixed kernel.

Table 3: Ablation studies on $\eta$ and $\eta_b$.

(a) PSNR ($\uparrow$).

| $\eta$ \ $\eta_b$ | 0.7 | 0.8 | 0.9 | 1.0 |
|---|---|---|---|---|
| 0.7 | 25.16 | 25.19 | 25.20 | 25.20 |
| 0.8 | 25.17 | 25.23 | 25.27 | 25.29 |
| 0.9 | 25.07 | 25.18 | 25.26 | 25.32 |
| 1.0 | 24.54 | 25.75 | 24.91 | 25.04 |

(b) KID $\times 10^3$ ($\downarrow$).

| $\eta$ \ $\eta_b$ | 0.7 | 0.8 | 0.9 | 1.0 |
|---|---|---|---|---|
| 0.7 | 16.27 | 14.30 | 12.76 | 11.65 |
| 0.8 | 21.07 | 19.07 | 17.37 | 15.98 |
| 0.9 | 27.85 | 25.64 | 23.81 | 22.40 |
| 1.0 | 45.10 | 42.50 | 40.10 | 37.84 |

# E    Ablation Studies on Hyperparameters

$\eta$ **and** $\eta_{\mathbf{b}}$.    Apart from the timestep schedules, DDRM has two hyperparameters $\eta$ and $\eta_b$, which control the level of noise injected at each timestep. To identify an ideal combination, we perform a hyperparameter search over $\eta, \eta_b \in \{0.7, 0.8, 0.9, 1.0\}$ for the task of deblurring with $\sigma_y = 0.05$ in 1000 ImageNet validation images, using the model trained in [13]. It is possible to also consider different $\eta$ values for $s_i = 0$ and $\sigma_i < \sigma_{\mathbf{y}}/s_i$; we leave that as future work.

We report PSNR and KID results in Table 3. From the results, we observe that generally (*i*) as $\eta_b$ increases, PSNR increases while KID decreases, which is reasonable given that we wish to leverage the information from $\mathbf{y}$; (*ii*) as $\eta$ increases, PSNR increases (except for $\eta = 1.0$) yet KID also increases, which presents a trade-off in reconstruction error and image quality (known as the perception-distortion trade-off [6]). Therefore, we choose $\eta_b = 1$ and $\eta = 0.85$ to balance performance on PSNR and KID when we report results.

**Timestep schedules.**    The timestep schedule has a direct impact on NFEs, as the wall-clock time is roughly linear with respect to NFEs [45]. In Tables 5 and 6, we compare the PSNR, FID, and KID of DDRM with 20 or 100 timesteps (with or without conditioning) and default $\eta = 0.85$ and $\eta_b = 1$. We observe that DDRM with 20 or 100 timesteps have similar performance when other hyperparameters are identical, with DDRM (20) having a slight edge in FID and KID.

# F    Experimental Setup of DGP, RED, and SNIPS

Recall that we evaluated DGP [38], RED [40], and SNIPS [25] on $256 \times 256$ ImageNet 1K images, for the problems of $4\times$ super resolution and deblurring without any noise in the measurements. Below we expand on the experimental setup of each one.

For DGP [38], we use the same hyperparameters introduced in the original paper for MSE-biased super resolution. We note that the downscaling applied in DGP is different from the block averaging filter that we used, and the numbers they reported are on the $128 \times 128$ resolution. Nevertheless, in our experiments, DGP achieved a PSNR of 23.06 on ImageNet 1K $256 \times 256$ block averaging $4\times$ super resolution, which is similar to the 23.30 reported in the original work. When applied on the deblurring problem, we retained the same DGP hyperparameters as well.

For RED [40], we apply the iterative algorithm only in the luminance channel of the image in the YCbCr space, as done in the original paper for deblurring and super resolution. As for the denoising engine enabling the algorithm, we use the same diffusion model used in DDRM to enable as fair a comparison as possible. We use the last step of the diffusion model (equivalent to denoising with $\sigma = 0.005$), as we found it to work best empirically. We also chose the steepest-descent version (RED-SD), and $\lambda = 500$ for best PSNR performance given the denoiser we used. We also set $\sigma_0 = 0.01$ when the measurements are noiseless, because $\sigma_0$ cannot be 0 as RED divides by it.

In super resolution, RED is initialized with the bicubic upsampled low-res image. In deblurring, it is initialized with the blurry image. We then run RED on the ImageNet 1K for different numbers of steps (see Table 4), and choose the best PSNR for each problem. Namely, we show in our paper RED on super resolution with 100 steps, and on deblurring with 500 steps. Interestingly, RED achieves a PSNR close to its best for super resolution in just 20 steps. However, DDRM (with 20 steps) still

Table 4: RED results on ImageNet 1K ($256 \times 256$) for $4\times$ super resolution and deblurring for different numbers of steps.

| | SUPER-RES | | DEBLURRING | |
|---|---|---|---|---|
| STEPS | PSNR↑ | KID↓ | PSNR↑ | KID↓ |
| 0 | 25.65 | 44.90 | 19.26 | 38.00 |
| 20 | 26.05 | 52.51 | 23.49 | 21.99 |
| 100 | 26.08 | 53.55 | 25.00 | 26.09 |
| 500 | 26.00 | 54.19 | 26.16 | 21.21 |

Table 5: ImageNet 50K validation set ($256 \times 256$) results on $4\times$ super resolution with additive noise of $\sigma_{\mathbf{y}} = 0.05$.

| METHOD | PSNR↑ | FID↓ | KID↓ | NFEs↓ |
|---|---|---|---|---|
| BICUBIC | 22.65 | 64.24 | 50.56 | 0 |
| DDRM | 24.70 | 20.16 | 15.25 | 100 |
| DDRM-CC | **24.71** | 18.22 | 13.57 | 100 |
| DDRM | 24.29 | 17.88 | 13.18 | 20 |
| DDRM-CC | 24.30 | **15.92** | **11.47** | 20 |

outperforms RED in PSNR, with substantially better perceptual quality (see Table 1 in the main paper).

Another interesting plug-and-play image restoration method is DPIR [57], which has recently achieved impressive results. It does so by applying the well-known Half Quadratic Splitting (HQS) plug-and-play algorithm using a newly proposed architecture. HQS requires an analytical solution of a minimization problem which is infeasible in general, due to the high memory requirements. DPIR provides efficient solutions for the specific degradation matrices $\mathbf{H}$ considered (circulant blurring, bicubic downsampling), which are different from the ones we consider (zero-padded blurring, block downsampling). In order to draw a fair comparison between the algorithms, one would have to use the same denosier architecture in both (as we have done for RED and SNIPS), and use the same degradation models. To apply DPIR on the same problems that we consider, we would need to substantially modify it and introduce efficient solutions. Therefore, we instead compare to RED, an alternative plug-and-play method.

SNIPS [25] did not originally work with ImageNet images. However, considering the method's similarity to DDRM (as both operate in the spectral space of $\mathbf{H}$), a comparison is necessary. We apply SNIPS with the same underlying diffusion model (with all 1000 timesteps) as DDRM for fairness. SNIPS evaluates the diffusion model $\tau$ times for each timestep. We set $\tau = 1$ so that SNIPS' runtime remains reasonable in comparison to the rest of the considered methods, and do not explore higher values of $\tau$. It is worth mentioning that in the original work, $\tau$ was set to 3 for an LSUN bedrooms diffusion model with 1086 timesteps. We set $c = 0.67$ as it achieved the best PSNR performance.

The original work in SNIPS calculates the SVD of $\mathbf{H}$ directly, which hinders its ability to handle $256 \times 256$ images on typical hardware. In order to draw comparisons, we replaced the direct calculation of the SVD with our efficient implementation detailed in Appendix D.

In Figure 4 and Table 2 in the main paper, we show that DGP, RED, and SNIPS all fail to produce viable results when significant noise is added to the measurements. For these results, we use the same hyperparameters used in the noiseless case for all algorithms (except $\sigma_{\mathbf{y}}$ where applicable). While tuning the hyperparameters may boost performance, we do not explore that option as we are only interested in algorithms where given $\mathbf{H}$ and $\sigma_{\mathbf{y}}$, the restoration process is automatic. To further demonstrate DDRM's capabilities and speed, we evaluate it on the entire $50,000$-image ImageNet validation set in Tables 5 and 6, reporting Fréchet Inception distance (FID) [18] as well as KID, as enough samples are available.

Table 6: ImageNet 50K validation set ($256 \times 256$) results on deblurring with additive noise of $\sigma_{\mathbf{y}} = 0.05$.

| METHOD | PSNR↑ | FID↓ | KID↓ | NFEs↓ |
|---|---|---|---|---|
| BLURRY | 18.05 | 93.36 | 74.13 | **0** |
| DDRM | 24.23 | 22.30 | 16.23 | 100 |
| DDRM-CC | 24.21 | 20.06 | 14.20 | 100 |
| DDRM | 24.60 | 21.60 | 15.65 | 20 |
| DDRM-CC | **24.61** | **19.66** | **13.94** | 20 |

## G   Runtime of Algorithms

In the main paper, we show the number of neural function evaluations (NFEs) as a proxy for the runtime of algorithms. Here, we consider the case of noisy deblurring, and measure the runtime of DDRM, RED, SNIPS, and DGP on an Nvidia RTX 3080 GPU. For each image, DDRM, RED, and SNIPS all run at around 0.09 s/it (seconds per iteration), with negligible differences of $< 0.01$s/it. We note that the denoiser model of DDRM, SNIPS, and RED is the same, so runtime is almost perfectly linearly correlated with NFEs. As for DGP, it uses a different model (a GAN), and it is slightly slower than our denoiser (0.11 s/it); this is partly because DGP requires additional gradient computations in order to perform an update. All in all, we observe that the runtime is indeed linear with NFEs, and since no algorithm has a significant runtime advantage over the rest, we prefer to use NFEs as a proxy for runtime, as it is a hardware-independent measure.

In this paper, we used pretrained generative models for image restoration. Since we didn't train any models, a single Nvidia RTX 3080 GPU was sufficient to run all experiments that were shown in the paper and the appendices.

## H   ILVR as a special case of DDRM

Given a generative diffusion model (*e.g.* DDPM [19]) that can predict $\mathbf{x}$ given $\mathbf{x}_{t+1}$ and $t+1$ for $t \in [0, T-1]$, and a noiseless measurement $\mathbf{y} = \boldsymbol{H}\mathbf{x}$, where $\boldsymbol{H}$ is a downscaling matrix, the Iterative Latent Variable Refinement (ILVR) [9] algorithm can sample from the posterior distribution $p_\theta^{(t)}(\mathbf{x}_t|\mathbf{x}_{t+1}, \mathbf{y})$ for $t \in [0, T-1]$.

We assume a variance exploding diffusion model, *i.e.* $\mathbf{x}_t = \mathbf{x} + \sigma_t \epsilon_t$ where $\epsilon_t \sim \mathcal{N}(0, \boldsymbol{I})$, without loss of generality (because it is equivalent to the variance preserving scheme, as we show in Appendix B). Under this setting, ILVR applies the following updates for $t = T-1, \dots, 0$:

$$\mathbf{x}'_t = \mathbf{x}_{\theta,t}(\mathbf{x}_{t+1}, t+1) + \sigma_t \epsilon_t,$$
$$\mathbf{y}_t = \boldsymbol{H}^\dagger \mathbf{y} + \sigma_t \epsilon'_t,$$
$$\mathbf{x}_t = \mathbf{x}'_t - \boldsymbol{H}^\dagger \boldsymbol{H} \mathbf{x}'_t + \boldsymbol{H}^\dagger \boldsymbol{H} \mathbf{y}_t,$$

where $\mathbf{x}_{\theta,t}(\mathbf{x}_{t+1}, t+1)$ is the prediction for $\mathbf{x}$ given by the diffusion model at timestep $t+1$, $\epsilon_t \sim \mathcal{N}(0, \boldsymbol{I})$, and $\epsilon'_t \sim \mathcal{N}(0, \boldsymbol{I})$. Substituting $\mathbf{x}'_t$, $\mathbf{y}_t$, and $\boldsymbol{H} = \boldsymbol{U\Sigma V}^T$, the last equation becomes

$$\begin{aligned}
\mathbf{x}_t &= \mathbf{x}'_t - \boldsymbol{H}^\dagger \boldsymbol{H} \mathbf{x}'_t + \boldsymbol{H}^\dagger \boldsymbol{H} \mathbf{y}_t \\
&= \mathbf{x}_{\theta,t}(\mathbf{x}_{t+1}, t+1) + \sigma_t \epsilon_t - \boldsymbol{H}^\dagger \boldsymbol{H} \left( \mathbf{x}_{\theta,t}(\mathbf{x}_{t+1}, t+1) + \sigma_t \epsilon_t \right) + \boldsymbol{H}^\dagger \boldsymbol{H} \left( \boldsymbol{H}^\dagger \mathbf{y} + \sigma_t \epsilon'_t \right) \\
&= \mathbf{x}_{\theta,t}(\mathbf{x}_{t+1}, t+1) + \sigma_t \epsilon_t - \boldsymbol{V\Sigma}^\dagger \boldsymbol{U}^T \boldsymbol{U\Sigma V}^T \left( \mathbf{x}_{\theta,t}(\mathbf{x}_{t+1}, t+1) + \sigma_t \epsilon_t \right) + \boldsymbol{V\Sigma}^\dagger \boldsymbol{U}^T \boldsymbol{U\Sigma V}^T \left( \boldsymbol{V\Sigma}^\dagger \boldsymbol{U}^T \mathbf{y} + \sigma_t \epsilon'_t \right) \\
&= \mathbf{x}_{\theta,t}(\mathbf{x}_{t+1}, t+1) + \sigma_t \epsilon_t - \boldsymbol{V\Sigma}^\dagger \boldsymbol{\Sigma V}^T \left( \mathbf{x}_{\theta,t}(\mathbf{x}_{t+1}, t+1) + \sigma_t \epsilon_t \right) + \boldsymbol{V\Sigma}^\dagger \boldsymbol{\Sigma V}^T \left( \boldsymbol{V\Sigma}^\dagger \boldsymbol{U}^T \mathbf{y} + \sigma_t \epsilon'_t \right) \\
&= \mathbf{x}_{\theta,t}(\mathbf{x}_{t+1}, t+1) + \sigma_t \epsilon_t - \boldsymbol{\Sigma}^\dagger \boldsymbol{\Sigma V V}^T \left( \mathbf{x}_{\theta,t}(\mathbf{x}_{t+1}, t+1) + \sigma_t \epsilon_t \right) + \boldsymbol{\Sigma}^\dagger \boldsymbol{\Sigma V V}^T \left( \boldsymbol{V\Sigma}^\dagger \boldsymbol{U}^T \mathbf{y} + \sigma_t \epsilon'_t \right) \\
&= \mathbf{x}_{\theta,t}(\mathbf{x}_{t+1}, t+1) + \sigma_t \epsilon_t - \boldsymbol{\Sigma}^\dagger \boldsymbol{\Sigma} \left( \mathbf{x}_{\theta,t}(\mathbf{x}_{t+1}, t+1) + \sigma_t \epsilon_t \right) + \boldsymbol{\Sigma}^\dagger \boldsymbol{\Sigma} \left( \boldsymbol{V\Sigma}^\dagger \boldsymbol{U}^T \mathbf{y} + \sigma_t \epsilon'_t \right).
\end{aligned}$$

The second to last equality holds because $\boldsymbol{\Sigma}^\dagger \boldsymbol{\Sigma}$ is a square diagonal matrix, and matrix multiplication with a square diagonal matrix is commutative. Recall that $\bar{\mathbf{x}}_t = \boldsymbol{V}^T \mathbf{x}_t$, $\bar{\mathbf{y}} = \boldsymbol{\Sigma}^\dagger \boldsymbol{U}^T \mathbf{y}$, and $\bar{\mathbf{x}}_{\theta,t} =$

$\boldsymbol{V}^T \mathbf{x}_{\theta,t}(\mathbf{x}_{t+1}, t+1)$, thus

$$
\begin{aligned}
\bar{\mathbf{x}}_t &= \boldsymbol{V}^T \mathbf{x}_{\theta,t}(\mathbf{x}_{t+1}, t+1) + \sigma_t \boldsymbol{V}^T \epsilon_t - \boldsymbol{V}^T \boldsymbol{\Sigma}^\dagger \boldsymbol{\Sigma} \left( \mathbf{x}_{\theta,t}(\mathbf{x}_{t+1}, t+1) + \sigma_t \epsilon_t \right) + \boldsymbol{V}^T \boldsymbol{\Sigma}^\dagger \boldsymbol{\Sigma} \left( \boldsymbol{V} \boldsymbol{\Sigma}^\dagger \boldsymbol{U}^T \mathbf{y} + \sigma_t \epsilon_t' \right) \\
&= \boldsymbol{V}^T \mathbf{x}_{\theta,t}(\mathbf{x}_{t+1}, t+1) + \sigma_t \boldsymbol{V}^T \epsilon_t - \boldsymbol{\Sigma}^\dagger \boldsymbol{\Sigma} \boldsymbol{V}^T \left( \mathbf{x}_{\theta,t}(\mathbf{x}_{t+1}, t+1) + \sigma_t \epsilon_t \right) + \boldsymbol{\Sigma}^\dagger \boldsymbol{\Sigma} \boldsymbol{V}^T \left( \boldsymbol{V} \boldsymbol{\Sigma}^\dagger \boldsymbol{U}^T \mathbf{y} + \sigma_t \epsilon_t' \right) \\
&= \boldsymbol{V}^T \mathbf{x}_{\theta,t}(\mathbf{x}_{t+1}, t+1) + \sigma_t \boldsymbol{V}^T \epsilon_t - \boldsymbol{\Sigma}^\dagger \boldsymbol{\Sigma} \boldsymbol{V}^T \mathbf{x}_{\theta,t}(\mathbf{x}_{t+1}, t+1) - \sigma_t \boldsymbol{\Sigma}^\dagger \boldsymbol{\Sigma} \boldsymbol{V}^T \epsilon_t + \boldsymbol{\Sigma}^\dagger \boldsymbol{\Sigma} \boldsymbol{V}^T \boldsymbol{V} \boldsymbol{\Sigma}^\dagger \boldsymbol{U}^T \mathbf{y} + \sigma_t \boldsymbol{\Sigma}^\dagger \boldsymbol{\Sigma} \boldsymbol{V}^T \epsilon_t' \\
&= \bar{\mathbf{x}}_{\theta,t} + \sigma_t \boldsymbol{V}^T \epsilon_t - \boldsymbol{\Sigma}^\dagger \boldsymbol{\Sigma} \bar{\mathbf{x}}_{\theta,t} - \sigma_t \boldsymbol{\Sigma}^\dagger \boldsymbol{\Sigma} \boldsymbol{V}^T \epsilon_t + \boldsymbol{\Sigma}^\dagger \boldsymbol{\Sigma} \boldsymbol{\Sigma}^\dagger \boldsymbol{U}^T \mathbf{y} + \sigma_t \boldsymbol{\Sigma}^\dagger \boldsymbol{\Sigma} \boldsymbol{V}^T \epsilon_t' \\
&= \left( \boldsymbol{I} - \boldsymbol{\Sigma}^\dagger \boldsymbol{\Sigma} \right) \bar{\mathbf{x}}_{\theta,t} + \left( \boldsymbol{I} - \boldsymbol{\Sigma}^\dagger \boldsymbol{\Sigma} \right) \sigma_t \boldsymbol{V}^T \epsilon_t + \boldsymbol{\Sigma}^\dagger \boldsymbol{U}^T \mathbf{y} + \boldsymbol{\Sigma}^\dagger \boldsymbol{\Sigma} \sigma_t \boldsymbol{V}^T \epsilon_t' \\
&= \left( \boldsymbol{I} - \boldsymbol{\Sigma}^\dagger \boldsymbol{\Sigma} \right) \bar{\mathbf{x}}_{\theta,t} + \left( \boldsymbol{I} - \boldsymbol{\Sigma}^\dagger \boldsymbol{\Sigma} \right) \sigma_t \boldsymbol{V}^T \epsilon_t + \bar{\mathbf{y}} + \boldsymbol{\Sigma}^\dagger \boldsymbol{\Sigma} \sigma_t \boldsymbol{V}^T \epsilon_t'.
\end{aligned}
$$

The matrix $\boldsymbol{\Sigma}^\dagger \boldsymbol{\Sigma}$ is a square diagonal matrix with zeroes in its entries where the singular value is zero, and ones otherwise. In addition, $\boldsymbol{\Sigma}^\dagger$ has a row of zeroes when the singular value is zero. Therefore, it holds that

$$
\bar{\mathbf{x}}_t^{(i)} = \begin{cases} \bar{\mathbf{x}}_{\theta,t}^{(i)} + \left( \sigma_t \boldsymbol{V}^T \epsilon_t \right)^{(i)} & \text{if } s_i = 0 \\ \bar{\mathbf{y}}^{(i)} + \left( \sigma_t \boldsymbol{V}^T \epsilon_t' \right)^{(i)} & \text{if } s_i \neq 0 \end{cases}, \tag{28}
$$

which in turn implies

$$
p_\theta^{(t)}(\bar{\mathbf{x}}_t^{(i)} | \mathbf{x}_{t+1}, y) = \begin{cases} \mathcal{N} \left( \bar{\mathbf{x}}_{\theta,t}^{(i)}, \sigma_t^2 \boldsymbol{I} \right) & \text{if } s_i = 0 \\ \mathcal{N} \left( \bar{\mathbf{y}}^{(i)}, \sigma_t^2 \boldsymbol{I} \right) & \text{if } s_i \neq 0 \end{cases}. \tag{29}
$$

This distribution is exactly the same as Equation 8 in the main paper when $\eta = \eta_b = 1$ and $\sigma_\mathbf{y} = 0$.

As for $\mathbf{x}_T$, ILVR initializes it by sampling from $\mathcal{N}\left(0, \sigma_T^2 \boldsymbol{I}\right)$ (or $\mathcal{N}\left(0, \boldsymbol{I}\right)$ in the variance preserving case) while DDRM samples according to Equation 7 in the main paper. The two initializations have the same variance but differ in the mean. This difference has a negligible effect on the end result since the variance is much larger than the difference in the means. Therefore, the above form of ILVR is a specific form of a DDRM (with $\eta = \eta_b = 1$), posed as a solution for linear inverse problems without noise in the measurements.

In their experiments, ILVR only tested $\boldsymbol{H}$ which is the bicubic downscaling matrix with varying scale factors. In theory, ILVR can also work for any linear degradation $\boldsymbol{H}$, as long as $\mathbf{y}$ does not contain noise.

# I   Additional Results

We provide additional figures below showing DDRM's versatility across different datasets, inverse problems, and noise levels (Figures 7, 8, 10, 11, and 12). We also showcase the sample diversity provided by DDRM in Figure 9; we present more uncurated samples from the ImageNet experiments in Figures 13 and 14. Moreover, we further illustrate DDRM's advantage over previous unsupervised methods by evaluating on two additional inverse problems: (i) $4\times$ super-resolution with the popular bicubic downsampling kernel; and (ii) deblurring with an anisotropic Gaussian blur kernel (with $\sigma = 20$ horizontally and $\sigma = 1$ vertically), mimicking motion blur. We show both noiseless and noisy versions in Tables 7 and 8, respectively. To maintain the unsupervised nature of the tested methods, we use the same hyperparameters as in block-averaging super-resolution and uniform deblurring.

Table 7: Noiseless $4\times$ super-resolution (using a bicubic kernel) and anisotropic Gaussian deblurring results on ImageNet 1K ($256 \times 256$).

| Method | $4\times$ super-resolution (Bicubic) | | | | Deblurring (Anisotropic) | | | |
|---|---|---|---|---|---|---|---|---|
| | PSNR↑ | SSIM↑ | KID↓ | NFEs↓ | PSNR↑ | SSIM↑ | KID↓ | NFEs↓ |
| Baseline | 26.06 | 0.73 | 72.41 | **0** | 19.96 | 0.58 | 25.23 | **0** |
| DGP | 20.82 | 0.50 | 29.62 | 1500 | 23.35 | 0.59 | 20.10 | 1500 |
| RED | 26.14 | 0.73 | 47.61 | 100 | 29.39 | 0.86 | 10.49 | 500 |
| SNIPS | 17.65 | 0.23 | 30.30 | 1000 | 33.34 | 0.86 | 0.58 | 1000 |
| DDRM | **27.09** | **0.76** | **12.78** | 20 | **36.02** | **0.93** | **0.41** | 20 |

Table 8: Noisy ($\sigma_{\mathbf{y}} = 0.05$) $4\times$ super-resolution (using a bicubic kernel) and anisotropic Gaussian deblurring results on ImageNet 1K ($256 \times 256$).

| Method | $4\times$ super-resolution (Bicubic) | | | | Deblurring (Anisotropic) | | | |
|---|---|---|---|---|---|---|---|---|
| | PSNR↑ | SSIM↑ | KID↓ | NFEs↓ | PSNR↑ | SSIM↑ | KID↓ | NFEs↓ |
| Baseline | 21.68 | 0.40 | 73.87 | **0** | 19.96 | 0.27 | 55.00 | **0** |
| DGP | 19.68 | 0.40 | 44.07 | 1500 | 22.64 | 0.53 | 25.38 | 1500 |
| RED | 22.65 | 0.46 | 54.90 | 100 | 11.97 | 0.10 | 130.30 | 500 |
| SNIPS | 16.16 | 0.14 | 69.69 | 1000 | 17.49 | 0.20 | 48.37 | 1000 |
| DDRM | **25.53** | **0.68** | **14.57** | 20 | **26.95** | **0.73** | **10.34** | 20 |

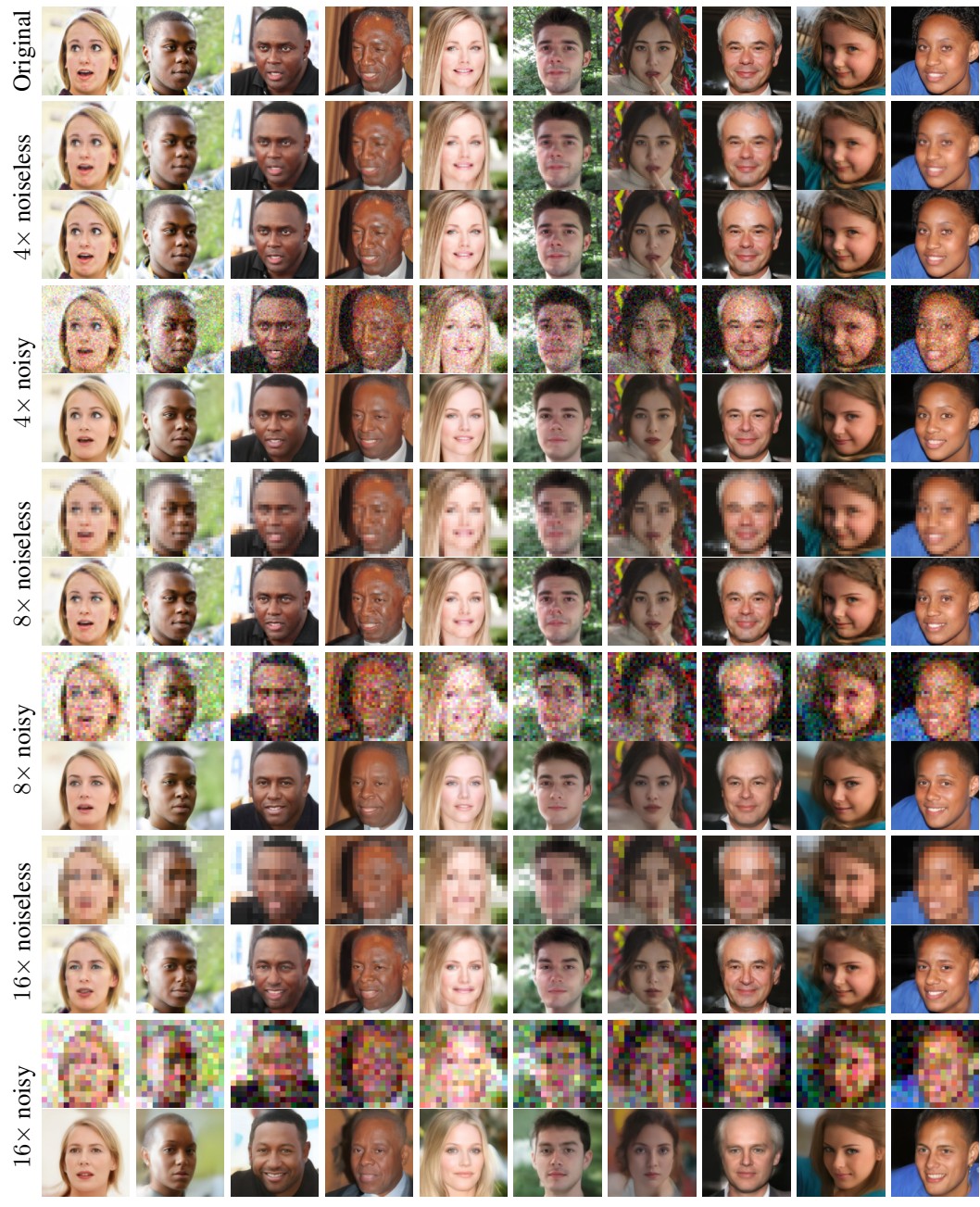

Figure 7: Pairs of low-res and recovered $256 \times 256$ face images with a 20-step DDRM. Noisy low-res images contain noise with a standard deviation of $\sigma_{\mathbf{y}} = 0.1$.

Original          Inpainting          Deblurring

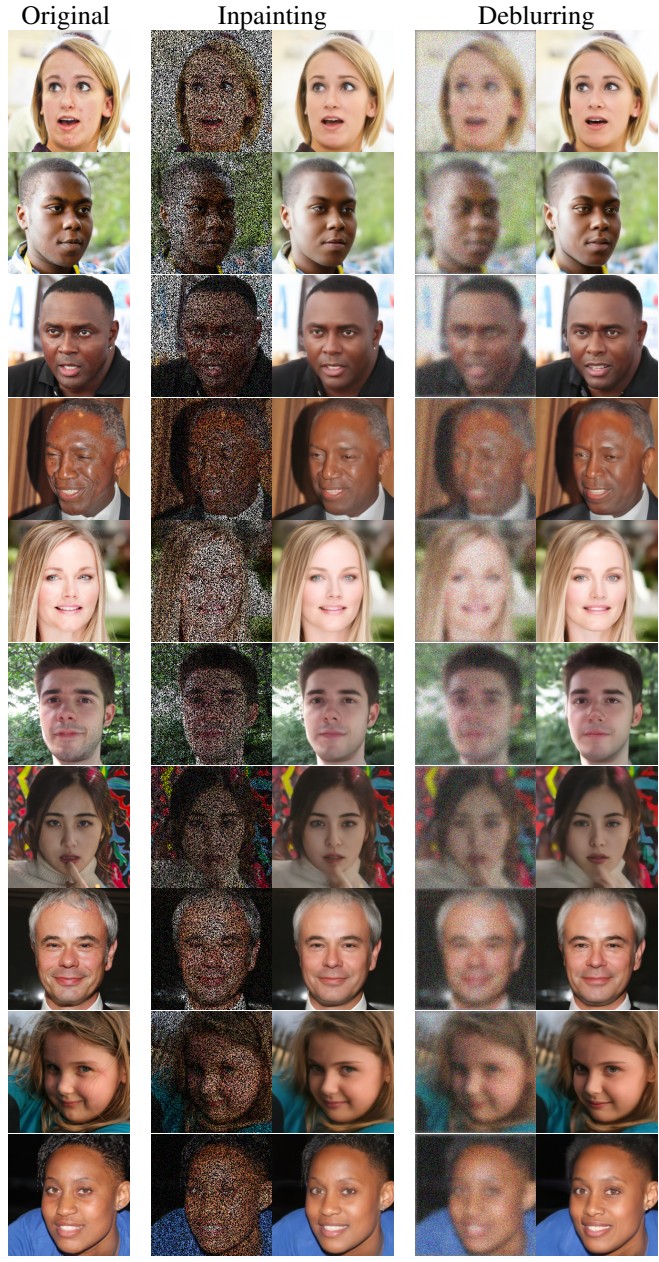

Figure 8: Pairs of degraded and recovered $256 \times 256$ face images with a 20-step DDRM. Degraded images contain noise with a standard deviation of $\sigma_\mathbf{y} = 0.1$.

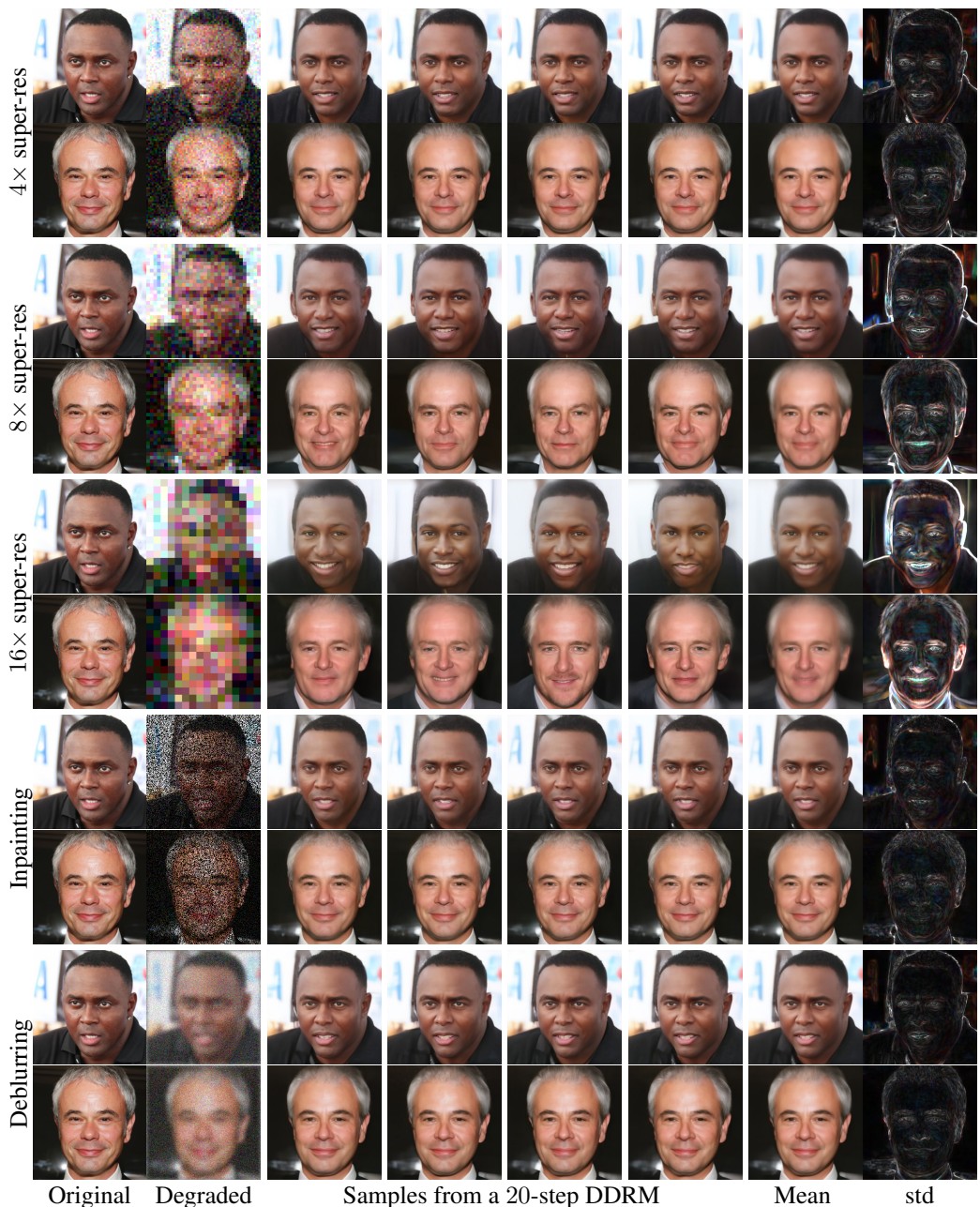

Original    Degraded       Samples from a 20-step DDRM       Mean    std

Figure 9: Original, degraded, and 6 recovered $256 \times 256$ face images with a 20-step DDRM. Degraded images contain noise with a standard deviation of $\sigma_{\mathbf{y}} = 0.1$. The mean and standard deviation (scaled by 4) of the sampled solution is shown.

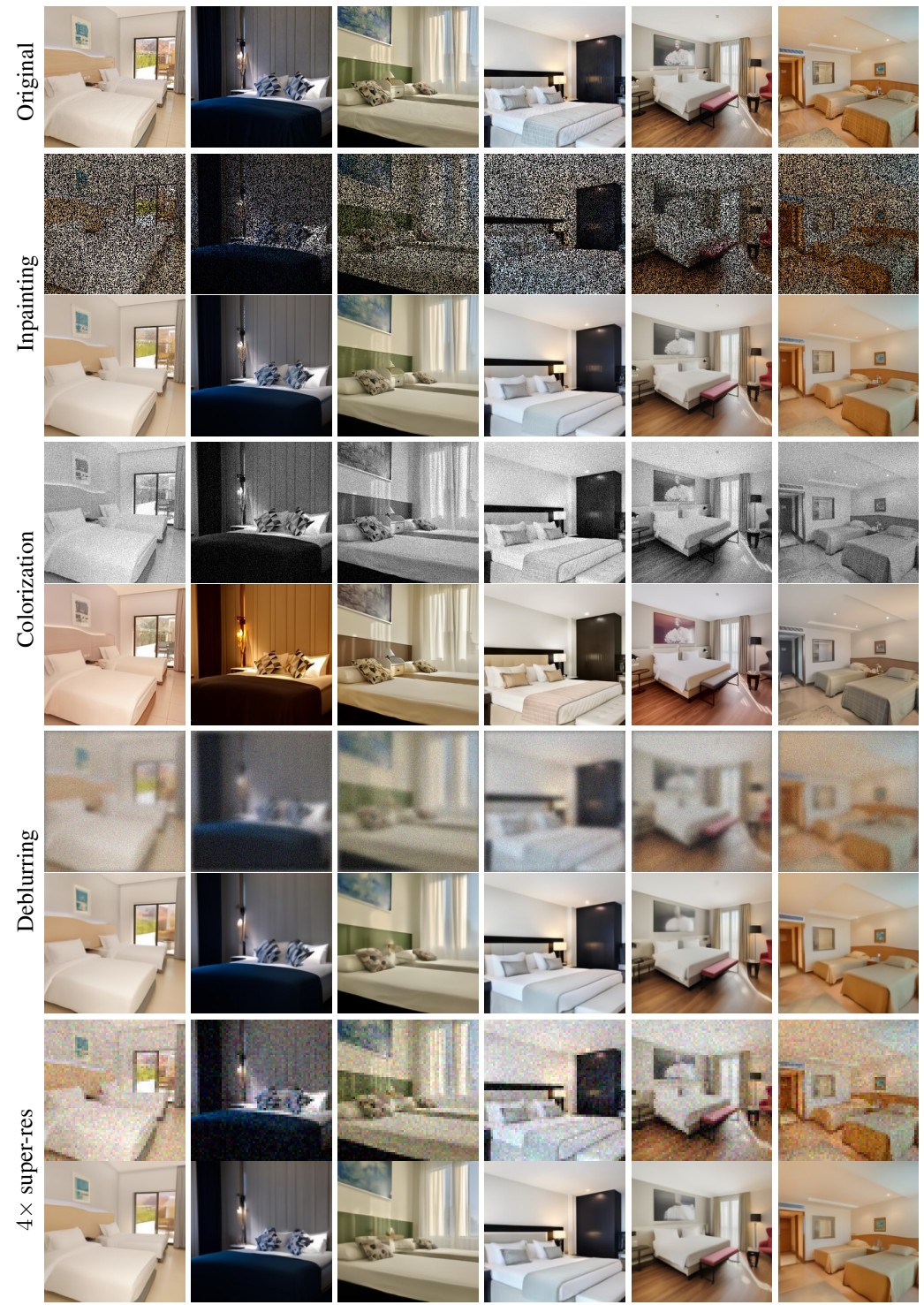

Figure 10: Pairs of degraded and recovered $256 \times 256$ bedroom images with a 20-step DDRM. Degraded images contain noise with a standard deviation of $\sigma_{\mathbf{y}} = 0.05$.

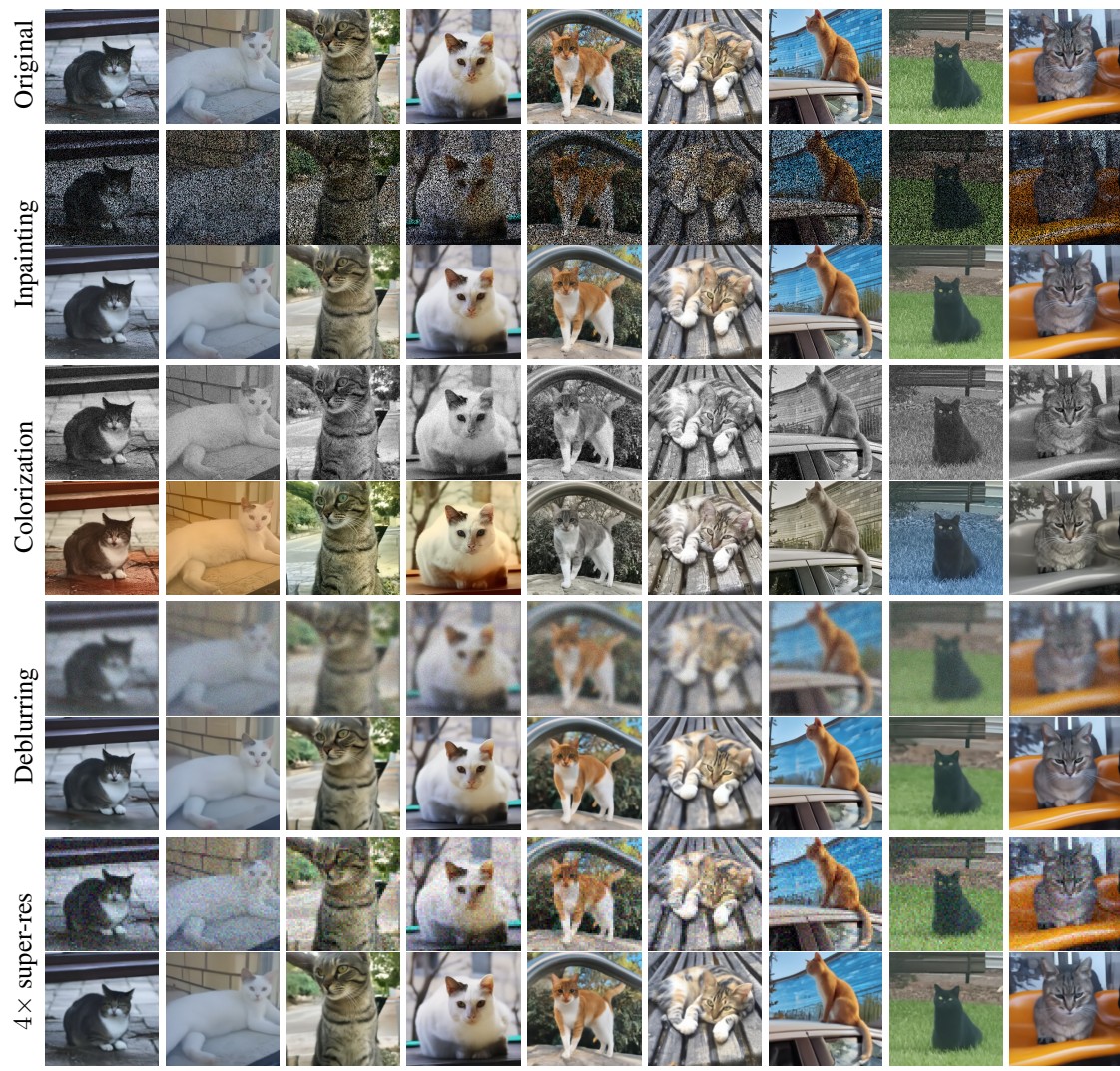

Figure 11: Pairs of degraded and recovered $256 \times 256$ cat images with a 20-step DDRM. Degraded images contain noise with a standard deviation of $\sigma_{\mathbf{y}} = 0.05$.

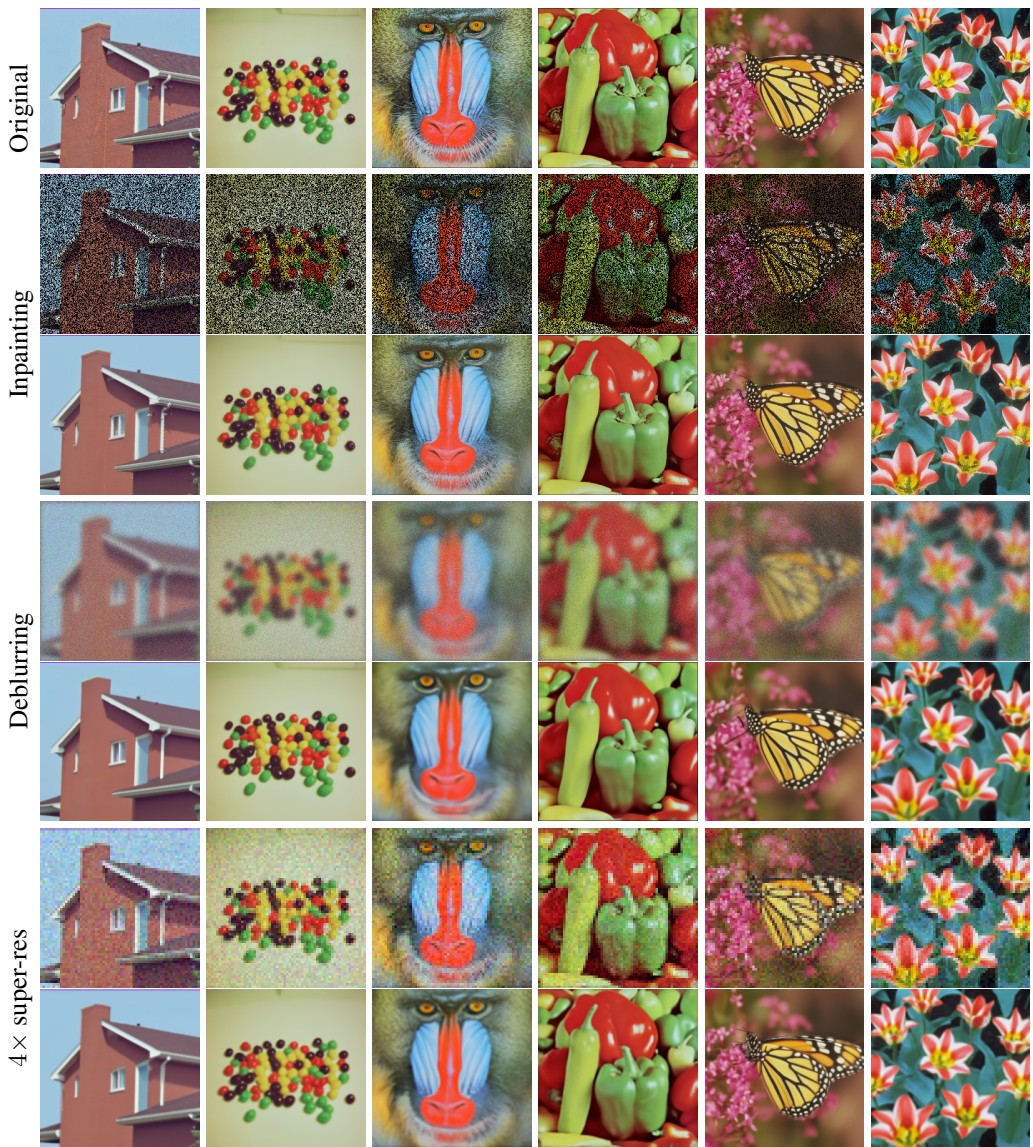

Figure 12: Pairs of degraded and recovered $256 \times 256$ USC-SIPI images with a 20-step DDRM using an ImageNet diffusion model. Degraded images contain noise with a standard deviation of $\sigma_{\mathbf{y}} = 0.05$.

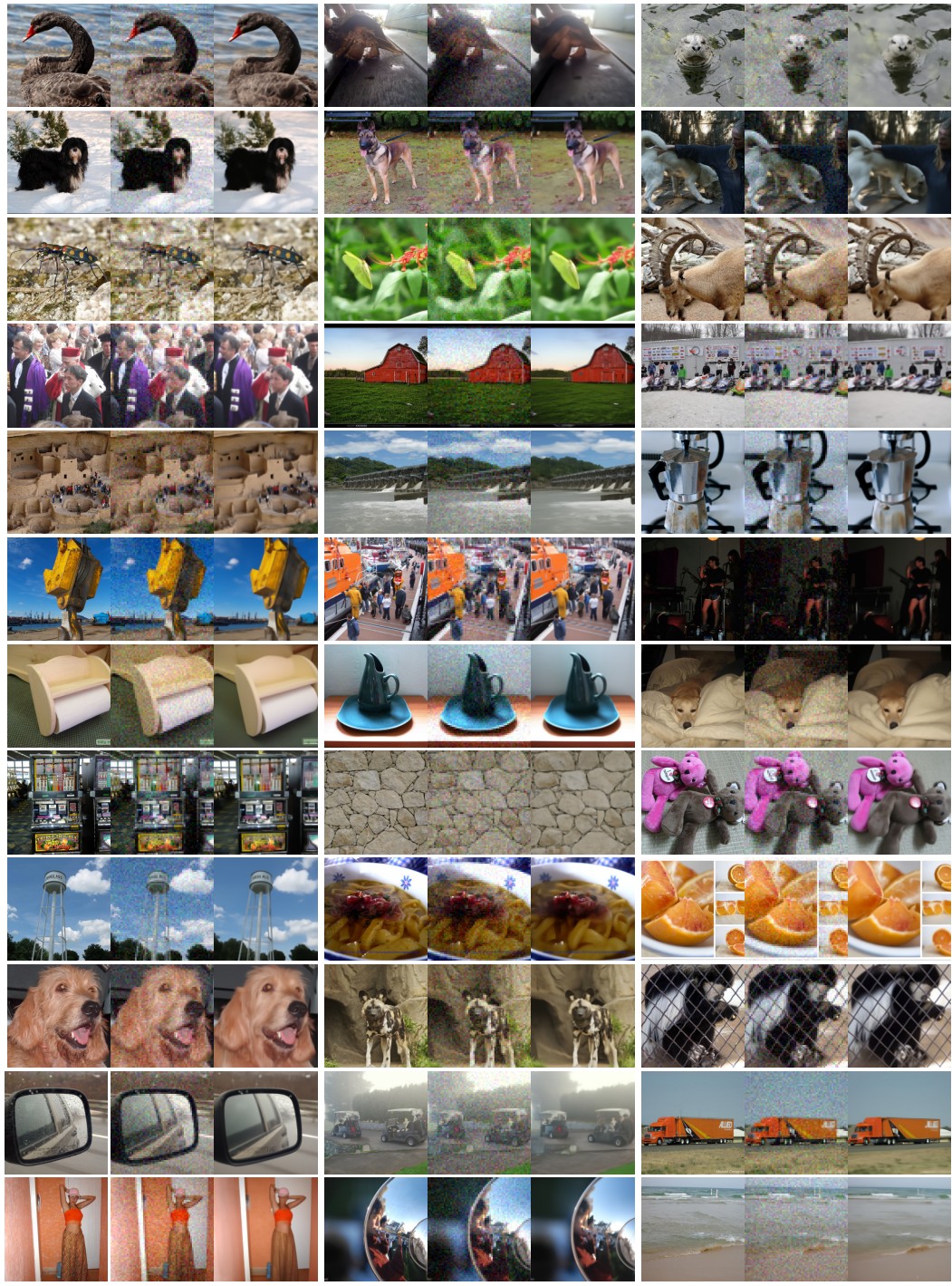

Figure 13: Uncurated samples from the noisy $4\times$ super resolution ($\sigma_{\mathbf{y}} = 0.05$) task on $256 \times 256$ ImageNet 1K. Each triplet contains (from left to right): the original image, the low-res image, and the restored image with DDRM-20.

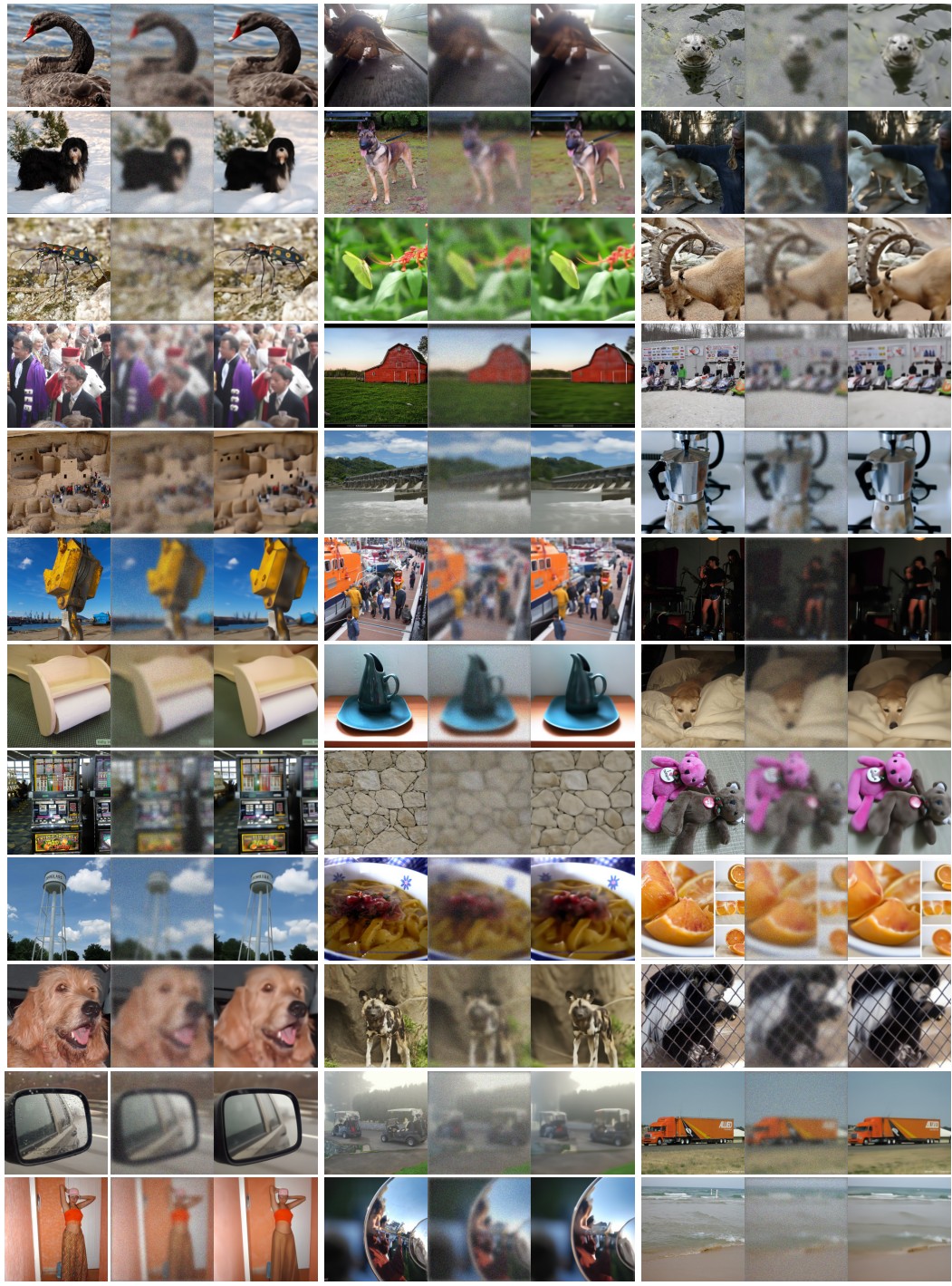

Figure 14: Uncurated samples from the noisy deblurring ($\sigma_{\mathbf{y}} = 0.05$) task on $256 \times 256$ ImageNet 1K. Each triplet contains (from left to right): the original image, the blurry image, and the restored image with DDRM-20.

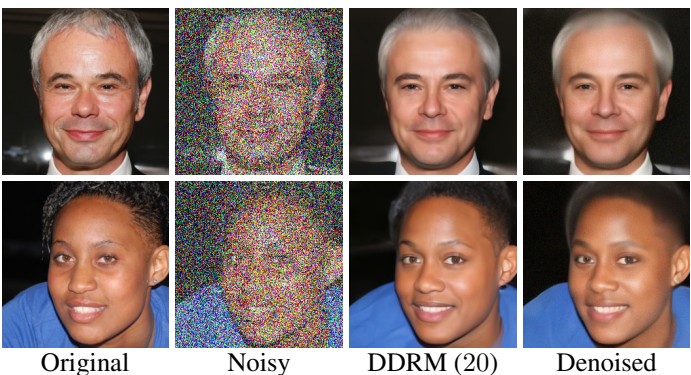

| Original | Noisy | DDRM (20) | Denoised |

Figure 15: Denoising ($\sigma_{\mathbf{y}} = 0.75$) face images. DDRM restores more fine details (*e.g.* hair) than an MMSE denoiser. The denoiser used here is the denoising diffusion function $f_\theta(\mathbf{x}_t, t)$ used by DDRM, where $t$ minimizes $|\sigma_t - \sigma_{\mathbf{y}}|$.