# OpenReview forum: "Denoising Diffusion Restoration Models"
_NeurIPS.cc/2022/Conference — NeurIPS 2022 Accept_

### Official Review · Reviewer_VSry · 2022-06-23

**Rating:** 6
**Confidence:** 4
**Soundness:** 4 excellent
**Presentation:** 4 excellent
**Contribution:** 4 excellent

**Summary:**

This paper studies image restoration problem and develops an efficient solution called denoising diffusion restoration model (DDRM). The key idea is to borrow principles from variational inference and exploit a pre-trained denoising diffusion generative model. This is conceptually similar to the popular pre-trained models in high-level vision (i.e., one pre-trained model supports different down-stream tasks). The DDRM has been tested on several image restoration tasks including SR, deblurring, inpainting, and colorization. The fast running speed is one of the salient features of the proposed DDRM.

**Questions:**

1. When DDRM produce a sequence of samples (restored images) like SNIPS, can you estimate their MSE like SURE does? If Yes, it might be possible to obtain a weighted (instead of unweighted) average in Fig. 4 of Supplementary Material.
2.  What assumption do you make about the degradation model H? Since Eq. (3) does SVD on H, H cannot be arbitrary, right? From Fig. 6, the current deblurring experiments assume optical blur. Can DDRM be applied to motion blur (but still satisfy the separable condition in D.5 of Supplementary)?
3. I am a bit confused by Fig. 1 in the Supplementary Material. What is the difference between "DDRM (20)" and "Denoised"? I assume "Denoised" means an MMSE denoiser. But the text can be more clear. Meanwhile, does DDRM reach a fixed-point iteration - i.e., x=f(x), where f is the denoiser? The motivation here is that we might assume clean images do satisfy such fixed-point condition.

**Ethics Review Area:**

["I don’t know"]

**Strengths And Weaknesses:**

Strengths:
+ The paper is well written and easy to follow.
+ DDRM is appealing because it makes little assumption about the degradation model (i.e., unsupervised approach).
+ Computational efficiency of DDRM is attractive for real-world applications.
+ DDRM has achieved improved restoration performance over other competing approaches including DGP (ECCV'2020) and SNIPS (NeurIPS'2021)

Weaknesses:
- It seems to me that the convergence issue of the DDRM is not discussed in the paper. Will DDRM-based image restoration always converge after 20 or 100 steps?

---

> ### Author Response · Authors · 2022-08-01
> **Author response**
>
> **Convergence issue:** DDRM follows a fixed noise schedule dictated by the schedule that the pre-trained DDPM has trained on. When choosing the number of steps, we effectively apply the 1000-step original schedule, but we skip some steps. For example, for a 100-step DDRM process, we skip 10 steps in each iteration ($t = 999, 989, \dots, 19, 9, 0$). DDRM converges with any given timestep budget, with the budget only affecting performance, not convergence.
>
> **MSE estimation:** This is indeed a very interesting question. Estimating the error similarly to SURE and providing a weighted average can be beneficial in some cases. However, we consider this out-of-scope for the current work, as we are focused on sampling from the posterior distribution.
>
> **Degradation model assumptions and deblurring:** We do not make any assumptions on the degradation model $H$, other than it being a linear operator. The SVD decomposition can be applied on any arbitrary matrix. Regarding motion blur, we opted not to show experiments as different motions induce different blurring kernels. However, as an approximation to the real-world motion blur problem, we conduct an additional experiment with an anisotropic Gaussian blur kernel (std = 20 and 1 horizontally and vertically, respectively) on ImageNet-1K, and obtain the following results:
>
> For noiseless deblurring, DDRM achieves 36.02 in PSNR and 0.41 in KID $\times 10^3$.
>
> For noisy deblurring ($\sigma_{y} = 0.05$), DDRM achievies 26.95 in PSNR and 10.34 in KID $\times 10^3$.
>
>
> **Terminology and fixed-point:** Thank you for bringing this terminology issue to our attention. You indeed assumed correctly. We will better clarify this in the final version of the paper.
> Regarding fixed-point, such a test is an interesting direction, which pertains mostly to the underlying DDPM denoiser model $f(x)$ rather than our image restoration technique, and therefore we consider this out-of-scope for this paper.

---

### Official Review · Reviewer_nQdA · 2022-07-04

**Rating:** 5
**Confidence:** 4
**Soundness:** 4 excellent
**Presentation:** 3 good
**Contribution:** 2 fair

**Summary:**

The paper proposes a method for the inverse problem based on diffusion model. Attributed to the unsupervised mechanism, it is able to
generalize well in several tasks as shown in the paper.

**Questions:**

See Weaknesses above.

**Limitations:**

This work is a non-blind method, and relies on the pre-provide degradation operator, which largely limits its practical value.

**Strengths And Weaknesses:**

Strengths:
1. As shown in Theorem C.2, the ELBO objective is equivalent to the form of that in DDPM/DDIM. Thus, the proposed method can readily employ the well-trained DDPM/DDIM model, which avoids the expensive training phase.

2. The proposed method is with fine generalization capability since it does not rely on specific degradation assumpution.

Weaknesses:
The main weakness of this work mainly focuses on the experiments. The experimental is too simple compared with related research works. Specifically,

1. In super-resolution, it only consider the block averaging filter. To the best of my knowledge, this is rarely used in super-resolution community.  Bicubic downsampling is the commonly used to evaluate the performance of different methods. I suggest the author should provide more experimental results under such setting, and compared with more recent methods.

2. For deblurring, aside from the more challenged motion blur case, the simple anisotropic Gaussian deblurring should be at least considered.

3. Even though announced to be left in future work, the proposed method relies on the pre-known degradation operator H, which is inaccessible in most of scenarios. Generally, this is a beautiful but unpractical work.

---

> ### Author Response · Authors · 2022-08-01
> **Author response**
>
> We are happy to provide more experimental evidence wherever possible, as per your request. Namely:
>
> **Bicubic super-resolution:** Thank you for your suggestion. We provide an additional experiment on ImageNet-1K with 4x bicubic downsampling, and the results are as follows:
>
> For noiseless SR, DDRM achieves 27.09 in PSNR and 12.78 in KID $\times 10^3$.
>
> For noisy SR ($\sigma_{y} = 0.05$), DDRM achievies 25.53 in PSNR and 14.57 in KID $\times 10^3$.
>
> We will also run comparisons with recent unsupervised methods for the final version of the paper. The results for bicubic SR are close to those of the block averaging SR, and we can expect similar behavior (maintaining performance level from block averaging SR) from other methods.
>
> **Other blurring kernels:** Motion blur is indeed more challenging as the blurring kernel is not exactly known, and thus we consider it out-of-scope for the paper. We thank you for your suggestion of anisotropic Gaussian blurring. Indeed, we conduct an experiment with an anisotropic Gaussian kernel (with std=20 and 1 in width and height, respectively) on ImageNet-1K, and obtain the following results:
>
> For noiseless deblurring, DDRM achieves 36.02 in PSNR and 0.41 in KID $\times 10^3$.
>
> For noisy deblurring ($\sigma_{y} = 0.05$), DDRM achievies 26.95 in PSNR and 10.34 in KID $\times 10^3$.
>
> We will compare these results with competing methods in the final version of the paper.
>
> **Unknown degradations:** There are several instances of inverse problems where the degradation model is known to the user at inference time, and DDRM provides a solution to a practical problem in those cases. Examples include inpainting (missing pixels are known), super resolution (most computer applications use standard algorithms for resizing images), compressed sensing (CT scans are conducted with known angles), and others. The generality of DDRM allows it to serve all these potential practical applications without the need for specialized training. Several successful previous works have attempted to solve inverse problems under similar settings [5, 34, 36, 45, 47].
> While there is interest in solving problems where the exact degradation operator is unknown, such scenarios necessarily require some assumptions about $H$ (e.g., blurring with an unknown kernel). We consider these scenarios out of scope for our work, as addressing them would depend on the assumptions made on $H$.
>
> We hope that the additional experiments provided in this response address your concerns regarding the experimental section.

---

> > ### Comment · Reviewer_nQdA · 2022-08-04
> > **Response to authors**
> >
> > 1. It would be better to provide the comparison results with other related methods on the Bicubic super-resolution and other blurring kernels.
> >
> > 2. Even though claimed out of scope, the proposed method cannot extend to the practical scenarios with unknow H.
> >
> > I still remain my rating.

---

### Official Review · Reviewer_u8iu · 2022-07-10

**Rating:** 5
**Confidence:** 4
**Soundness:** 3 good
**Presentation:** 3 good
**Contribution:** 3 good

**Summary:**

This paper proposes Denoising Diffusion Restoration Models (DDRM), which is the first general sampling-based inverse problem solver for various unsupervised image restoration tasks. DDRM outperforms existing baselines and is efficient in terms of NFEs. Also, DDRM is more beneficial when there is noise in the input images.

**Questions:**

- It is preferred to report quantitative results according to the number of iterations for the proposed method. What happens if more than 20 iterations are applied for the proposed method in Table1,2?

- Most of the experiments were performed on ImageNet. Does the proposed method work well for frequently used datasets for each task? For instance, Set5, Set14, B100, Urban100 and Manga109 for super-resolution. What happens to verify the proposed method by applying it to real-world data such as medical imaging mentioned in the introduction?


**Ethics Review Area:**

["I don’t know"]

**Limitations:**

- PSNR and KID were used as evaluation metrics. I am curious about quantitative results when other evaluation metrics are utilized for image reconstruction problems, such as BRISQUE, NIQE, PIQE, RankIQA, MetaIQA.

BRISQUE: Mittal, Anish, Anush Krishna Moorthy, and Alan Conrad Bovik. "No-reference image quality assessment in the spatial domain." IEEE Transactions on Image Processing (TIP) 21.12 (2012): 4695-4708.

NIQE: Mittal, Anish, Rajiv Soundararajan, and Alan C. Bovik. "Making a “completely blind” image quality analyzer." IEEE Signal Processing Letters (SPL) 20.3 (2012): 209-212.

PIQE: Venkatanath, N., et al. "Blind image quality evaluation using perception based features." 2015 Twenty First National Conference on Communications (NCC). IEEE, 2015.

RankIQA: Liu, Xialei, Joost Van De Weijer, and Andrew D. Bagdanov. "Rankiqa: Learning from rankings for no-reference image quality assessment." Proceedings of the IEEE International Conference on Computer Vision (ICCV). 2017.

MetaIQA: Zhu, Hancheng, et al. "MetaIQA: Deep meta-learning for no-reference image quality assessment." Proceedings of the IEEE/CVF Conference on Computer Vision and Pattern Recognition (CVPR). 2020.


**Strengths And Weaknesses:**

Strength

- The proposed method requires much fewer iterations compared to existing diffusion-based methods.

- The proposed method was experimented on various image restoration tasks including super-resolution, inpainting and colorization.

Weakness

- As mentioned in the Conclusions, the proposed method is focused on linear inverse problems, so there is likely to be a weakness in the non-linear inverse problem that exists in real data. It would be good to suggest the possible limitations and the analysis when the proposed method is applied to the non-linear inverse problem setup.

- Also, there are many cases where the degradation operator is unknown at the inference stage. Therefore, it would be good to have a mention on how to apply the proposed method in a practical real-world situation (degradation operator is unknown).

- Although the above two limits seem out-of-scope of this paper, it  would be good to analyze the proposed method from various angles by experimenting with various data and setups. For example, it is possible to consider various data sets, image resolution and quality, as well as non-linear inverse problem cases and unknown degradation operators.

---

> ### Author Response · Authors · 2022-08-01
> **Author response**
>
> **Non-linear degradations:** For non-linear degradations, our method cannot be applied as it requires a SVD decomposition of the degradation operator, which can only be applied on a linear degradation model. We will discuss this limitation in the final version of the paper. We therefore focus on the linear setting, as was done in several previous papers [5, 34, 36, 45, 47].
>
> **Unknown degradations at inference time**: There are several instances of inverse problems where the degradation model is known to the user at inference time, and DDRM provides a solution to a practical problem in those cases. Examples include inpainting (missing pixels are known), super resolution (most computer applications use standard algorithms for resizing images), compressed sensing (CT scans are conducted with known angles), and others. The generality of DDRM allows it to serve all these potential practical applications without the need for specialized training. Several successful previous works have attempted to solve inverse problems under similar settings [5, 34, 36, 45, 47].
> While there is interest in solving problems where the exact degradation operator is unknown, such scenarios necessarily require some assumptions about $H$ (e.g., blurring with an unknown kernel). We consider these scenarios out of scope for our work, as addressing them would depend on the assumptions made on $H$.
>
> **What happens if more iterations are used**: Thank you for your suggestion. We evaluate PSNR on a similar setting with 32 examples with different number of steps (50, 100, 200, 500, 1000), similar to what is discussed in the response to Reviewer ghAD.
>
> | Steps        | 20    | 50    | 100   | 200   | 500   | 1000  |
> |--------------|-------|-------|-------|-------|-------|-------|
> | SR4          | 26.05 | 26.35 | 26.42 | 26.54 | 26.56 | 26.42 |
> | Noisy SR4    | 24.97 | 25.20 | 25.26 | 25.37 | 25.39 | 25.21 |
> | Deblur       | 36.69 | 37.80 | 38.49 | 39.17 | 39.87 | 40.28 |
> | Noisy Deblur | 25.08 | 24.97 | 24.72 | 23.81 | 20.53 | 17.01 |
>
> From the results, it seems that increasing iterations results in only marginal improvements compared to the 20 steps that we use, except for noisy deblurring (which becomes worse, possibly because the hyperparameters are tuned for the 20-step case). We also note that this comes with spending 2.5x to 50x more compute compared to our default setting.
>
> **Consider more traditional datasets such as Set5**: We show an example for USC-SIPI images in Figure 6, where we perform noisy deblurring. This dataset is very frequently used for testing several image processing tasks, and has some images in common with Set14. Regarding medical applications, there were no publicly available diffusion models trained on medical data before the NeurIPS deadline, and we consider designing and training such a model out-of-scope.
>
> **More image quality metrics:** We currently evaluate image fidelity using KID, which is known to correlate well with human perception. We plan to add additional metrics in the final version.

---

> > ### Comment · Reviewer_u8iu · 2022-08-08
> > **Author Rebuttal Acknowledgement**
> >
> > Thank you for the response. The rebuttal is satisfactory to me, thus I keep my original rating.

---

### Official Review · Reviewer_ghAD · 2022-07-11

**Rating:** 6
**Confidence:** 3
**Soundness:** 3 good
**Presentation:** 3 good
**Contribution:** 3 good

**Summary:**

This paper addresses the efficiency problem of the unsupervised approach in image restoration. The authors proposed DDRM, a denoising diffusion generative model that uses variational inference to learn the the posterior distribution of a linear inverse problem for image restoration. Specifically, the authors define a diffusion process of image restoration using variational distributions, where the trainable parameters of the distributions can be learnt by maximizing an ELBO objective. The experiments on ImageNet show that DDRM has good performance and outperforms previous methods.

**Questions:**

(1) From the visual results, it seems that sometimes the results are oversmooth, e.g., see the hand in Figure 4. It would be nice to evaluate the output images with a metric on image structure similarity, e.g., SSIM. Could the authors consider adding this metric to Table 1 or 2?

(2) In Table 1 and 2, how many NFEs are required for DDRM to outperform previous methods? Could the authors also provide the runtime statistics (memory, time) for each NFE? These numbers might support the efficiency claim better.

(3) Section 3.3: from a practical point of view, what happens if we learn a different model for each inverse problem? Would that give even better performance?

**Limitations:**


- There are a few typos in the manuscript, e.g., delubrring, diffsuion. Please use a spell checker to double check.


**Strengths And Weaknesses:**

I like the paper in that it presents a solid and sound technical approach with clear theorems and proofs for using diffusion generative models for image restoration. The focus of the paper is on defining the variational objective and the relevant distributions in the diffusion process, which allows efficient inference. The contribution is novel and significant.

The results of the paper appear to be convincing enough. Both quantitative and qualitative results demonstrate strong performance compared to previous methods.

The writing is great. The paper reads well and relatively easy to understand.

On the downside, the experiments shown in the paper can be more extensive, and some analyses on the degradation model could be added. As the proposed method is claimed as an unsupervised method for image restoration with priors learnt on ImageNet, it is a pity that only synthetic degradation models are shown in the results. As ImageNet is a dataset with real images, it would be nice to understand further how well the real degradation can be handled with the proposed model.

---

> ### Author Response · Authors · 2022-08-01
> **Author response**
>
> **Real-world degradations**: There are several instances of inverse problems where the degradation model is known to the user at inference time, and DDRM provides a solution to a practical problem in those cases. Examples include inpainting (missing pixels are known), super resolution (most computer applications use standard algorithms for resizing images), compressed sensing (CT scans are conducted with known angles), and others. The generality of DDRM allows it to serve all these potential practical applications without the need for specialized training. Several successful previous works have attempted to solve inverse problems under similar settings [5, 34, 36, 45, 47].
> While there is interest in solving problems where the exact degradation operator is unknown, such scenarios necessarily require some assumptions about $H$ (e.g., blurring with an unknown kernel). We consider these scenarios out of scope for our work, as addressing them would depend on the assumptions made on $H$.
>
> **Report SSIM**: Thank you for the suggestion. We have obtained SSIM results for DDRM on ImageNet-1K as follows: 0.722 (SR4), 0.948 (Deblur), 0.662 (Noisy SR4), 0.663 (Noisy Deblur). We will add these results in the final version, alongside SSIM results for competing methods.
>
> **How DDRM performs in fewer number of steps**: We perform an additional experiment on 32 examples from ImageNet, where we change the number of steps (10, 11, 12, 14, 16, 20) but keep other hyperparameters intact.
>
> | Steps        | 10    | 11    | 12    | 14    | 16    | 20    |
> |--------------|-------|-------|-------|-------|-------|-------|
> | SR4          | 22.83 | 24.86 | 25.51 | 25.97 | 25.99 | 26.05 |
> | Noisy SR4    | 22.02 | 23.61 | 24.41 | 24.89 | 24.93 | 24.97 |
> | Deblur       | 35.68 | 35.83 | 35.95 | 36.21 | 36.38 | 36.69 |
> | Noisy Deblur | 24.95 | 24.96 | 24.95 | 25.04 | 25.01 | 25.08 |
>
> It appears that for SR4, 14-step DDRM is comparable to the 20-step version; once the number of steps falls below that, the PSNR drops rapidly. In deblurring, 10-step still gives decent results, which is expected as deblurring is simpler. We note that these results are close to the number of steps required in unconditional generation in efficient samplers such as DDIM.
>
> **Runtime comparisons**: We discuss these results in Appendix G in the supplementary material. The SVD initialization (e.g., for deblurring) takes 0.42 seconds on an RTX 3080 GPU. Afterwards, For each image, both DDRM and RED run at around 0.09 s/it (seconds per iteration), with a negligible difference of <0.01 s/it. Thus, especially when using larger batch sizes, most of the runtime comes from the denoiser, not the matrix operations thereafter or the initialization.
>
> We note that the denoiser model of DDRM, SNIPS, and RED is the same, so runtime is almost perfectly linearly correlated with NFEs. DGP uses a different model (a GAN), but it is slightly slower than our denoiser (0.11 s/it); this is partly because DGP requires additional gradient computations.
>
> **Training problem specific models**: Thank you for the suggestion. Problem-specific models (which lose generality) would be expected to perform better than DDRM. These are already suggested in works such as SR3 and Palette. However, the authors of the two papers did not release their code or ImageNet models and did not respond to our communications regarding this matter.
>
> As high-resolution diffusion models are very expensive to train (the pre-trained ADM model that we use is trained for over 100 V100-days), we believe that it is useful to consider methods that are not problem-specific, which would generalize to cases such as different resolution factors and different noise levels in the original data.
>
> **Typos**: Thank you for pointing these out. We will fix these and spellcheck in the final version.

---

> > ### Comment · Reviewer_ghAD · 2022-08-05
> > **Thank you for the responses**
> >
> > The rebuttal is almost satisfactory to me except that I haven't seen the SSIM for the competing methods. It is hard to conclude about the performance ranking with this metric for now.

---

### Author Response · Authors · 2022-08-01
**Thank you to the reviewers**

We thank the reviewers for acknowledging the novelty and technical contribution of our work. We are glad that the reviewers find that DDRM has “improved image restoration performance over other competing methods” from recent top-tier conferences (VSry), “requires much fewer iterations compared to existing diffusion-based methods” (u8iu), “avoids the expensive training phase” (nQDA), and is “relatively easy to understand” (ghAD).

We also thank the reviewers for their valuable and constructive feedback on our paper. In the following responses, we address the issues raised by the reviewers.

---

### Meta-Review · Area_Chair_nspW · 2022-08-25

**Recommendation:** Accept
**Confidence:** Less certain

**Metareview:**

In this paper, the authors present a variational inference approach based on a diffusion model to learning the posterior distribution of an unsupervised linear inverse problem.  The presented method is evaluated on multiple linear inverse problems and beats state of the art unsupervised inversion methods while running significantly faster than them.  The algorithm has strong theoretical support, in the form of a rigorously proven equivalence with unconditional denoising with diffusion models.  The paper could be improved by a more extensive set of experiments, including investigation of stability against incorrect degradation models.  Overall, the algorithmic advance with a strong theoretical foundation outweighs these weaknesses, and the paper is recommended for acceptance.

**Award:**

No

---

### Decision · Program_Chairs · 2022-09-14

Accept